# Multiple structures of RNA polymerase II isolated from human nuclei by ChIP-CryoEM analysis

Tomoya Kujirai [1,2], Junko Kato[1], Kyoka Yamamoto[1,3], Seiya Hirai[1,3], Takeru Fujii[4], Kazumitsu Maehara [4,7], Akihito Harada [4,5], Lumi Negishi [1], Mitsuo Ogasawara[1], Yuki Yamaguchi [6], Yasuyuki Ohkawa [4], Yoshimasa Takizawa [1] & Hitoshi Kurumizaka [1,2,3] ✉

RNA polymerase II (RNAPII) is a central transcription enzyme that exists as multiple forms with or without accessory factors, and transcribes the genomic DNA packaged in chromatin. To understand how RNAPII functions in the human genome, we isolate transcribing RNAPII complexes from human nuclei by chromatin immunopurification, and determine the cryo-electron microscopy structures of RNAPII elongation complexes (ECs) associated with genomic DNA in distinct forms, without or with the elongation factors SPT4/5, ELOF1, and SPT6. This ChIP-cryoEM method also reveals the two EC-nucleosome complexes corresponding nucleosome disassembly/reassembly processes. In the structure of EC-downstream nucleosome, EC paused at superhelical location (SHL) −5 in the nucleosome, suggesting that SHL(−5) pausing occurs in a sequence-independent manner during nucleosome disassembly. In the structure of the EC-upstream nucleosome, EC directly contacts the nucleosome through the nucleosomal DNA-RPB4/7 stalk and the H2A-H2B dimer-RPB2 wall interactions, suggesting that EC may be paused during nucleosome reassembly. These representative EC structures transcribing the human genome provide mechanistic insights into understanding RNAPII transcription on chromatin.

The genomic DNA regions that encode proteins and some non-coding RNAs are transcribed by RNA polymerase II (RNAPII) in eukaryotes[1,2]. RNAPII consists of twelve subunits, which are highly conserved from yeast to humans. During gene expression, RNAPII is first assembled on the promoter region of genes and forms a pre-initiation complex, followed by transcription initiation. The RNAPII then moves forward to the coding DNA region and elongates the RNA transcripts. When RNAPII reaches the termination region of the gene, it dissociates from the DNA. The dissociated RNAPII can be recycled for the next round of transcription[3].

In eukaryotic cells, genomic DNA is organized within the nucleus as chromatin, utilizing a set of four histones, H2A, H2B, H3, and H4. Two of

[1]Laboratory of Chromatin Structure and Function, Institute for Quantitative Biosciences, The University of Tokyo, 1-1-1 Yayoi, Bunkyo-ku, Tokyo, Japan. [2]Laboratory for Transcription Structural Biology, RIKEN Center for Biosystems Dynamics Research, 1-7-22 Suehiro-cho, Tsurumi-ku, Yokohama, Japan. [3]Department of Biological Sciences, Graduate School of Science, The University of Tokyo, 1-1-1 Yayoi, Bunkyo-ku, Tokyo, Japan. [4]Division of Transcriptomics, Medical Institute of Bioregulation, Kyushu University, 3-1-1 Maidashi, Higashi, Fukuoka, Japan. [5]Department of Multi-Omics, Graduate School of Medical Sciences, Kyushu University, 3-1-1 Maidashi, Higashi-ku, Fukuoka, Japan. [6]School of Life Science and Technology, Institute of Science Tokyo, 4259 Nagatsuta, Yokohama, Japan. [7]Present address: Department of Multi-Omics, Graduate School of Medical Sciences, Kyushu University, 3-1-1 Maidashi, Higashi-ku, Fukuoka, Japan. ✉e-mail: kurumizaka@iqb.u-tokyo.ac.jp

each histone associate as an octamer and form the core of a structure known as the nucleosome, around which 145–147 base pairs of DNA are wrapped around the histone octamer[4]. The nucleosomal DNA positions are defined as superhelical locations (SHLs) −7 through 0 to + 7, with about 10 base-pair periodicity. In chromatin, nucleosomes are interconnected by DNA segments referred to as linker DNA, creating an overall structure that visually resembles a "beads-on-a-string" conformation[5–7]. To perform vital genome functions, the transcription machinery must navigate the traversal of nucleosomes in chromatin.

In transcription elongation on chromatin, various transcription elongation factors are associated with the RNAPII elongation complex (EC). SPT4 and SPT5 form a complex called DSIF (DRB sensitivity-inducing factor) and are implicated in promoter-proximal pausing, productive transcription elongation, and termination regulation[8–11]. SPT6 binds the phosphorylated RNAPII C-terminal domain and the RPB4/7 stalk on the RNAPII, and is associated with active transcription, splicing, histone modifications, and transcription termination[12–17]. ELOF1 (the yeast Elf1 homolog) genetically interacts with other transcription elongation factors, including SPT4/5, SPT6, and TFIIS, and promotes elongation and transcription-coupled DNA repair[18–22].

In vitro transcription studies coupled with cryo-electron microscopy (cryo-EM) analyses have illuminated the step-by-step choreography of nucleosome transcription by *Komagataella pastoris* RNAPII and transcription elongation factors. During this process, RNAPII EC incrementally uncoils the nucleosomal DNA from the histone surface up to the position before the nucleosomal dyad (SHL(0)), aided by the basal transcription elongation factor, TFIIS, and substantially pauses at the SHL(−5) and SHL(−1) positions of the nucleosome[23]. Consistently, previous genomics studies revealed that EC pausing at the SHL(−5) and SHL(−1) positions is predominantly observed in the cells[24–26]. This process, including the sequential unwrapping of the nucleosomal DNA, is drastically potentiated in the presence of key transcription elongation factors, Spt4/5 and Elf1[19,27]. The synergy of additional transcription elongation factors, Spt6, Spn1, and Paf1C, together with Spt4/5 and Elf1, further enhances the transcription on the nucleosome by EC[28,29]. The nucleosomal histones are then relocated from the leading to the trailing side of the transcribing EC with the aid of these transcription elongation factors and a representative histone chaperone, FACT[29]. Notably, in the EC-nucleosome complexes paused near the SHL(−1) position, the trailing DNA region of the transcribing EC can be rewrapped with the histones located in front of EC[30,31]. These findings may reflect potential intermediates for an alternative histone relocation pathway with mammalian and *K. pastoris* ECs, in a histone chaperone-independent manner.

A genetic study with yeast identified at least 50 factors suggested to function in chromatin transcription, including the elongation factors Spt4, Spt5, and Spt6[32–34]. These ~ 50 factors commonly function in the suppression of cryptic transcription, which is usually repressed by chromatin structure, and are categorized into histones, histone gene regulators, histone chaperones, histone modifiers, nucleosome remodelers, transcription elongation factors, and Mediator components. These findings suggest that the integrity of gene expression by the suppression of undesired cryptic transcription is maintained by groups of factors involved in chromatin structure and dynamics.

To gain deeper insight into the architectural dynamics of RNAPII in human nuclei, we established the ChIP-CryoEM method, in which protein complexes associated with genomic DNA and/or chromatin are isolated by immunoprecipitation from human cells. We then determined the cryo-EM structures of native RNAPII ECs transcribing genomic DNA in human cells.

## Results

### ChIP-CryoEM reveals distinct structures of transcribing RNA polymerase II in human nuclei

To study the RNAPII structures during transcription in the human genome, we produced the FLAG-His$_6$-tagged RPB3, which is an essential subunit of human RNAPII, in HeLa cells[35], and performed the ChIP-CryoEM analysis (Supplementary Fig. 1a–d). HeLa cells are commonly used for biochemical and genomic analyses and suitable for the ChIP-CryoEM analysis, because they can be prepared in large scale suspension cultures. The HeLa cell chromatin in isolated nuclei was solubilized by sonication and micrococcal nuclease treatment (Fig. 1a). The RNAPII molecules in the soluble fraction were purified by anti-FLAG antibody beads under physiological salt conditions (150 mM NaCl). Mass spectrometric analysis and western blot analysis revealed the presence of preinitiation complex components, such as Mediator, TFIIH, and TFIID, elongation complex components, such as Serine 2 phosphorylation of RPB1 C-terminal domain, SPT4/5, SPT6, PAF1C, and ELOF1, and histones in this immunopurified fraction (Supplementary Fig. 1e, f and Supplementary Data 1). Therefore, various transcription stages of RNAPII molecules on chromatin can be captured by this method, although transcription factors weakly associated with RNAPII may dissociate during sample preparation. Note that a relatively small amount of PAF1 was associated with RNAPII, and RTF1, a subunit of PAF1C, was detected only when the sample was prepared under low salt conditions (50 mM) (Supplementary Fig. 1f). This weak PAF1C association with RNAPII may be consistent with the previous report that PAF1C is transiently associated with the RNAPII EC[36], and suggested that less stringent conditions may capture factors weakly associated with RNAPII. At the same time, the amounts of non-specific proteins were apparently increased in the eluted sample (Supplementary Fig. 1g), and the efficiency of RNAPII elution from the beads was largely decreased under the low salt conditions (Supplementary Fig. 1f, g). Therefore, the RNAPII sample eluted under the 150 mM NaCl conditions was chosen and then fractionated by sucrose gradient ultracentrifugation with glutaraldehyde fixation (GraFix) (Supplementary Fig. 1d). In parallel, the eluted sample was also fractionated by sucrose gradient ultracentrifugation using the same separation conditions without fixation, and the RNAPII-associated factors within the fractions were detected by western blot analysis (Fig. 1b and Supplementary Fig. 1h). We found that the majority of the RNAPII (FLAG) co-fractionated with serine 2 phosphorylation and transcription elongation factors, SPT5 and SPT6 (Fig. 1b, fractions 4-9). The amount of PAF1 did not correlate with those of SPT5 and SPT6. Meanwhile, the preinitiation complex components, MED17 and XPB, were mainly enriched in the bottom fraction (Fig. 1b, fraction 15), suggesting that these factors formed large complexes. We subjected the GraFix fractions 4-11, which may mainly contain the RNAPII elongation complex with SPT5 and SPT6 without aggregation, to the cryo-EM analysis (Supplementary Figs. 1d, 2–7 and Tables 1, 2).

We first obtained two distinct native RNAPII EC structures, EC with RNAPII alone and EC with RNAPII complexed with SPT4/5, ELOF1, and SPT6 at 2.7 Å and 3.1 Å resolutions, respectively (Fig. 1c, d). These ECs were actively transcribing genomic DNA or stalled on certain genomic regions in human cells. In both EC structures, all twelve protein subunits were observed (Fig. 1c, d). Genomic DNA and nascent RNA fragments were clearly visualized around the RNAPII catalytic center, suggesting that they were actively transcribing forms of human RNAPII ECs (Supplementary Fig. 8a, d). Since these are average structures of RNAPIIs transcribing various genomic DNA loci, the cryo-EM maps of the DNA and RNA bases also represent an average of heterogeneous sequences. Further, these ECs may be mixtures of pre-translocated and post-translocated forms because the densities of the RNA base at the +1 position are relatively weak (Supplementary Fig. 8b, c, e, f). ECs without elongation factors may be observed as a consequence of elongation factor dissociation during sample preparation for the cryo-EM analysis. Alternatively, ECs without elongation factors may function in genome transcription at certain loci. Further studies are required to solve this issue.

In the EC-SPT4/5-ELOF1-SPT6 structure, endogenous elongation factors, SPT4/5, ELOF1, and SPT6, are tightly associated with EC

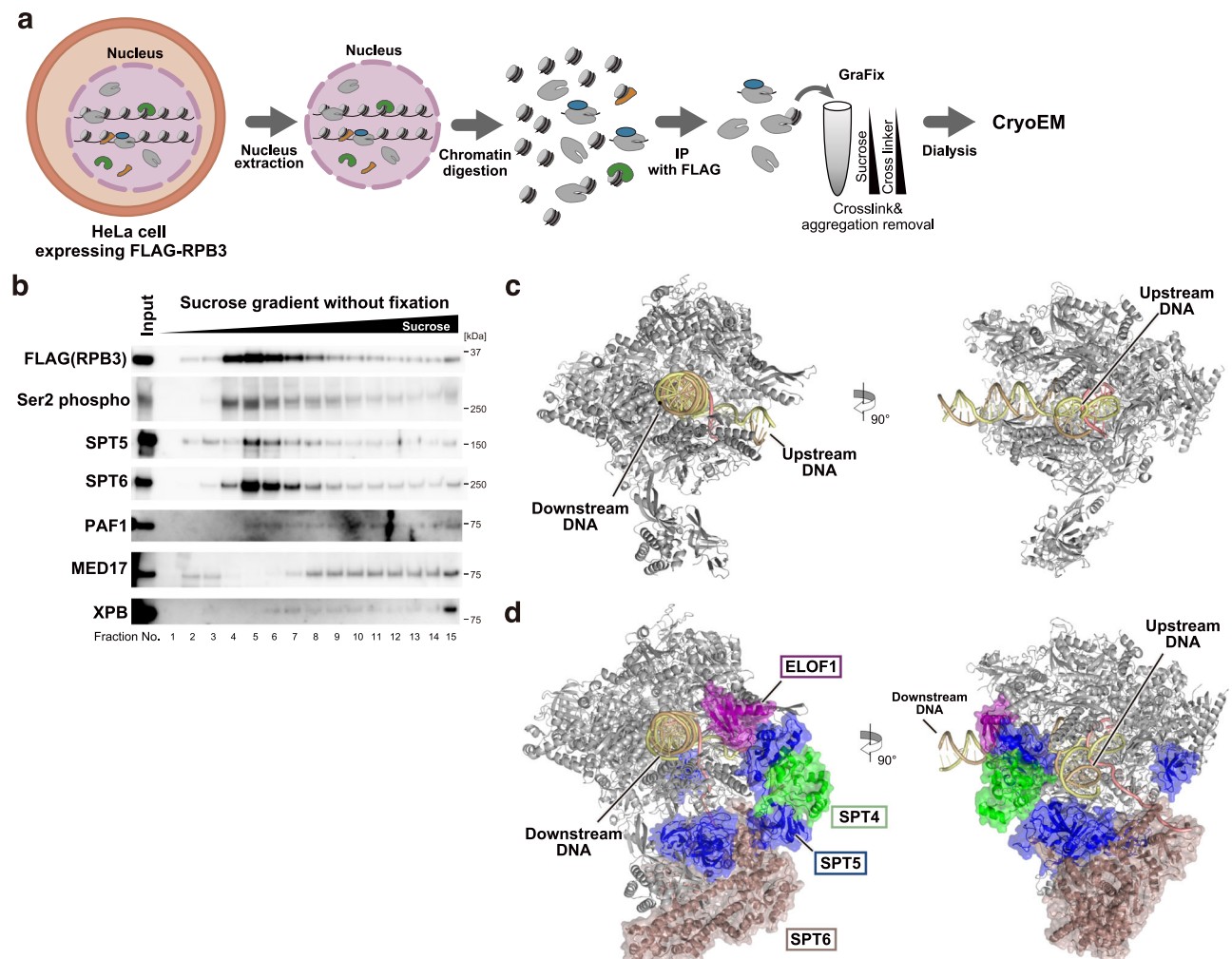

**Fig. 1 | Transcribing RNAPII EC structures in human nuclei revealed by ChIP-cryoEM. a** Schematic of the ChIP-cryoEM method. **b** Western blot of sucrose gradient experiment without crosslinker. The indicated factors were detected by antibodies. This experiment was repeated twice with similar results. Source data are provided as a Source Data file. **c** Cryo-EM structure of the EC extracted from HeLa cell nuclei (**d**) Cryo-EM structure of the transcribing EC extracted from HeLa cell nuclei.

## Table 1 | Cryo-EM data collection statistics

| Batch | batch1 | batch2 | batch3 | batch4 | batch5 |
|---|---|---|---|---|---|
| Microscope | Krios G4 | Krios G4 | Krios G4 | Krios G4 | Krios G4 |
| Voltage (kV) | 300 | 300 | 300 | 300 | 300 |
| Detector | K3/BioQuantum | K3/BioQuantum | K3/BioQuantum | K3/BioQuantum | K3/BioQuantum |
| Slit width (eV) | 20 | 20 | 20 | 20 | 20 |
| Magnification | 81,000 | 81,000 | 81,000 | 81,000 | 81,000 |
| Pixel size for data collection (Å) | 1.06 | 1.06 | 1.06 | 1.06 | 1.06 |
| Total electron exposure (e⁻/Å²) | 59.1 | 59.6 | 59.9 | 61.1 | 61.2 |
| Exposure time (s) | 4.5 | 4.5 | 4.5 | 4.5 | 4.5 |
| Exposure rate (e⁻/pixel/sec) | 13.1 | 13.2 | 13.3 | 13.6 | 13.6 |
| Number of frames | 40 | 40 | 40 | 40 | 40 |
| Defocus range (mm) | −1.0 to −2.5 | −1.0 to −2.5 | −1.0 to −2.5 | −1.0 to −2.5 | −1.0 to −2.5 |
| Number of collected micrographs | 594 | 6126 | 14143 | 3951 | 8161 |

(Fig. 1d). SPT4 consists of a globular domain, and SPT5 contains the NGN, KOW, and CTR domains (Fig. 2a)[8]. Amino acid residues 22 to 83 of ELOF1 were visualized as the globular domain, which is composed of three β-strands and an α-helix (Fig. 2a, b). Similar to yeast *K. pastoris* Elf1, human ELOF1 directly interacted with the SPT5 NGN domain,

RPB2 lobe, and RPB1 clamphead in the EC-SPT4/5-ELOF1-SPT6 structure, forming the DNA entry tunnel[10] (Figs. 1c and 2b). The SPT4 and SPT5 NGN-KOW1 domains were bound to RNAPII around the upstream DNA and formed a DNA exit tunnel (Figs. 1c and 2c). The SPT5 KOW2, KOW3, KOWx, and KOW4 domains were bound

**Table 2 | Cryo-EM reconstruction statistics**

| | EC | EC-SPT4/5-ELOF1-SPT6 | EC-downstream nucleosome | EC-upstream nucleosome |
|---|---|---|---|---|
| Image processing software | Relion | Relion | Relion | Relion |
| Number of particles | 523284 | 132687 | 21652 | 32065 |
| Pixel size for refinement (Å) | 1.06 | 1.06 | 1.06 | 1.06 |
| Symmetry imposed | C1 | C1 | C1 | C1 |
| Resolutions (Å) (FSC = 0.143) | | | | |
| Overall | 2.7 | 3.1 | 4.1 | 3.6 |
| Part | – | 3.5 (stalk-SPT6) | 4.3 (nucleosome) | 3.6 (nucleosome) |
| Part | – | – | 3.7 (RNAPII) | 3.3 (RNAPII) |
| **Model composition** | | | | |
| Protein residues | 3886 | 5341 | 4616 | 4588 |
| RNA/DNA residues | 73 | 98 | 325 | 316 |
| Ligands | 8 Zn, 1 Mg | 10 Zn, 1 Mg | 8 Zn, 1 Mg | 8 Zn, 1 Mg |
| **Model refinement** | | | | |
| Refinement package | Phenix (realspace refine) | Phenix (realspace refine) | Phenix (realspace refine) | Phenix (realspace refine) |
| CCvolume/CCmask | 0.85/0.86 | 0.84/0.85 | 0.56/0.57 | 0.74/0.75 |
| MolProbity score | 1.02 | 1.15 | 1.29 | 1.38 |
| Clash score | 1.10 | 1.50 | 3.10 | 4.07 |
| RMSDs | | | | |
| Bond length (Å) | 0.008 | 0.008 | 0.009 | 0.007 |
| Bond angle (°) | 1.458 | 1.125 | 1.490 | 1.197 |
| Ramachandran plot (%) | | | | |
| Outliers | 0.00 | 0.00 | 0.00 | 0.00 |
| Allowed | 3.27 | 3.81 | 3.16 | 3.14 |
| Favored | 96.73 | 96.19 | 96.84 | 96.86 |
| Rotamer outliers (%) | 0.00 | 0.04 | 0.00 | 0.02 |
| PDB ID | 8XSO | 8XRM | 8XVS | 8XRJ |
| EMDB ID | EMD-38624 | EMD-38607 | EMD-38717 | EMD-38604 |

the RPB4/7 stalk, RPB1 clampcore, RPB1 dock, and RPB2 clamp (Fig. 2d). KOW5 was bound within a pocket formed by RPB1 dock, RPB2 wall, RPB3, RPB11, and RPB12, forming the RNA exit tunnel (Fig. 2e). These results are consistent with the positions reported in reconstituted RNAPII elongation complexes with recombinant elongation factors[10,14,37].

SPT6 directly interacted with SPT5, the RPB4/7 stalk, and the exit RNA (Fig. 1d). Human SPT6 is composed of 1,726 amino acid residues, and contains the N-terminal region, core region, and tSH2 domain (Fig. 2a)[17]. In our EC-SPT4/5-ELOF1-SPT6 structure, the SPT6 core region and the N-terminal region close to the core were visualized. The SPT6 areas containing the N-terminal, Cradle, YqgF, HhH, and S1 regions formed the SPT5 binding surface and tightly associated with the KOW1, KOW2, KOW3, KOWx, and KOW4 domains of SPT5 (Fig. 2f–i). The SPT6 N-terminal region formed a bridge between the SPT5 KOW1 and KOW3 domains (Fig. 2f). The SPT6 HhH and YqgF domains interacted with the SPT5 KOW1 domain (Fig. 2g). The SPT6 YqgF, S1, and Cradle domains bound the SPT5 KOWx and KOW4 domains (Fig. 2h, i). The binding of SPT6 follows a similar mode to that of recombinant yeast Spt6 in the reconstituted EC-Spt4/5-Elf1-Paf1C-Spt6-Spn1 structures[29] (Fig. 2f–i and Supplementary Fig. 9). This observation is also consistent with the recently published structure of the reconstituted mammalian EC-SPT4/5-PAF1C-SPT6-IWS1-TFIIS-SETD2 complex[38]. However, in these EC structures, including our own, SPT6 binds more closely to the SPT5 KOW1 and KOW3 domains compared to the previously reported reconstituted mammalian EC-SPT4/5-PAF1C-SPT6 structures[14,39]. This difference may reflect the inherent flexibility of SPT6 binding.

## The structure of EC paused in the downstream nucleosome in human nuclei

We next determined the structure of an EC associated with a downstream nucleosome. In this complex, EC paused in the downstream nucleosome, at approximately 20 base pairs (the SHL(−5) position) from the entry DNA site (Fig. 3a, b). Elongation factors, such as SPT4/5, ELOF1, and SPT6, were not detected in this EC. The nucleosomal DNA region is peeled by EC until the DNA position is bound to the H2A L2-loop, in which the H2A Arg77 residue may capture the DNA backbone (Fig. 3b, c). Consequently, approximately 20 base pairs of the nucleosomal DNA (the SHL(-7)-SHL(-6) DNA region) are detached from the histone surface (Fig. 3b). In addition, the nucleosomal SHL(+0.5)-SHL(+2) DNA region directly contacts the RPB2 lobe and RPB1 clamphead (Fig. 3d). These histone H2A-DNA and EC-nucleosomal DNA interactions may be responsible for the EC pausing at the SHL(−5) position. The structure of this human EC-nucleosome complex is remarkably similar to that of the reconstituted *K. pastoris* EC-nucleosome complex, in which EC is strikingly paused at the SHL(−5) position as revealed by in vitro transcription experiments[23,40] (Supplementary Fig. 10). Interestingly, previous genome-wide studies have consistently revealed that EC is predominantly paused around the nucleosomal SHL(−5) position in cells[25,26]. The existence of this complex suggests that the SHL(-5) pausing may commonly occur in a DNA sequence-independent manner during chromatin transcription.

A previous cryo-EM structure of the *K. pastoris* EC-Spt4/5-Elf1-nucleosome complex revealed that Spt4/5 and Elf1 sterically intervene in the direct interaction between the EC and the nucleosome at the SHL(+0.5)-SHL(+2) DNA region and alleviate the SHL(-5) pausing[19]. As

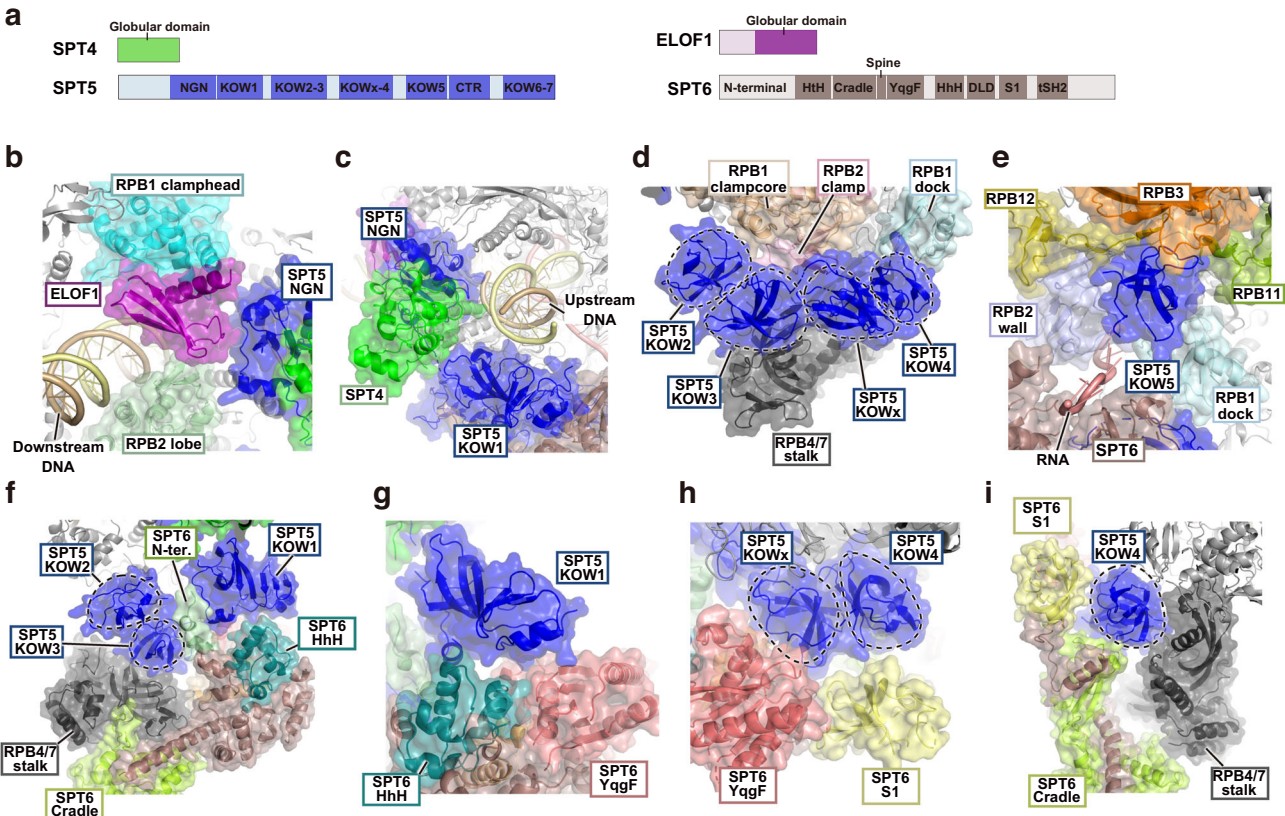

**Fig. 2 | Details of the EC-SPT4/5-ELOF1-SPT6 structure. a** Domain architectures of the elongation factors. The known domains are indicated. **b** Close-up view of ELOF1 (purple). **c** Close-up view around the upstream DNA. **d** Close-up view of the SPT5 KOW2, KOW3, KOWx, and KOW4 domains. Each KOW domain is indicated with dotted circles. SPT6 is omitted for clarity. **e** Close-up view of the KOW5 domain. **f** The SPT6-SPT5 KOW3-RPB4/7 stalk interaction. **g** The SPT5 KOW1-SPT6 HhH and YqgF interaction. **h** Close-up view of the SPT5 KOWx and KOW4-SPT6 YqgF and S1 domain interactions. The RNA is omitted for clarity. **i** The SPT5 KOW4-SPT6 S1 and Cradle interactions. ELOF1, RPB1 clamphead, RPB1 dock, RPB2 lobe, RPB4/7 stalk, SPT5, SPT4, and SPT6 are shown as ribbon models overlaid with transparent surface models.

described above, in the EC-downstream nucleosome complex reported here, there is no space for ELOF1 binding on the EC-nucleosome interface, due to the direct contact of EC with the nucleosomal DNA (Fig. 3e). However, the elongation factors, such as SPT4/5 and ELOF1, may have dissociated from the EC during sample preparation and the RNAPII-nucleosome interactions observed in this structure may have arisen from a change in the nucleosome positions after the SPT4/5 and ELOF1 dissociation. A single-molecule study revealed that yeast Spt4/5 dynamically associates and dissociates with the EC molecule in the elongation step[41]. Therefore, the RNAPII without elongation factors may be a transcribing RNAPII form before SPT4/5 loading, and alternatively, the SPT4/5 and ELOF1 dissociation during sample preparation may occur because of the dynamic nature of SPT4/5. Further study is required to clarify this issue.

**The EC structure complexed with the upstream nucleosome in human nuclei**

We next determined another EC-nucleosome complex structure, in which the nucleosome sits on the upstream DNA region behind EC (Fig. 4a). This could be a reassembled state of the nucleosome after EC has traversed it. In this complex, EC did not contain the elongation factors. Approximately 120 base pairs (SHL(+ 4.5)-SHL(−7) DNA region) of the upstream DNA were wrapped around the histone octamer (Fig. 4b). In this complex, we observed the EM density maps corresponding to the canonical H2A- and H3-specific residues, such as H2A Tyr39, H2A Arg99, and H3 Met90 (Supplementary Fig. 11). Accordingly,

this structure may contain some of the canonical histones H2A and H3.1 or H3.2. However, the present structure may be an averaged structure with heterogeneous nucleosomes containing various histone variants. Therefore, the small residues specific to histone variants, such as Thr (for H2A.Z.1), Pro (for MacroH2A1.2), and Gly (for H2A.X, MacroH2A1.2, and H3.3), may not be visualized if they exist, because the EM densities from the larger side chains of canonical histones may bury the densities of those residues.

In the EC-upstream nucleosome structure, the H2A-H2B dimer located close to EC was partially exposed on the nucleosome surface and directly contacts the RPB2 wall of EC (Fig. 4c). In addition, the RPB1 clampcore and RPB4/7 stalk hold the SHL(-1)-SHL(-3) DNA region of the nucleosome (Fig. 4d). There was no space for SPT5 and SPT6 binding on the EC-nucleosome interface, due to the direct contact of EC with the upstream nucleosome (Fig. 4e). As mentioned above, it is also possible that these direct contacts may be formed after SPT4/5 and ELOF1 dissociation during sample preparation. Given the existence of this EC-upstream nucleosome complex in human nuclei, EC may be paused during the nucleosome reassembly process, even though the downstream nucleosome has been disassembled in advance and is not present. The interactions of EC with the upstream reassembled nucleosome may promote EC pausing during and/or immediately after nucleosome reassembly.

We then compared the present EC-upstream nucleosome structure with the in vitro reconstituted structure of the yeast *K. pastoris* EC with transcription elongation factors (EC115), in which the nucleosome

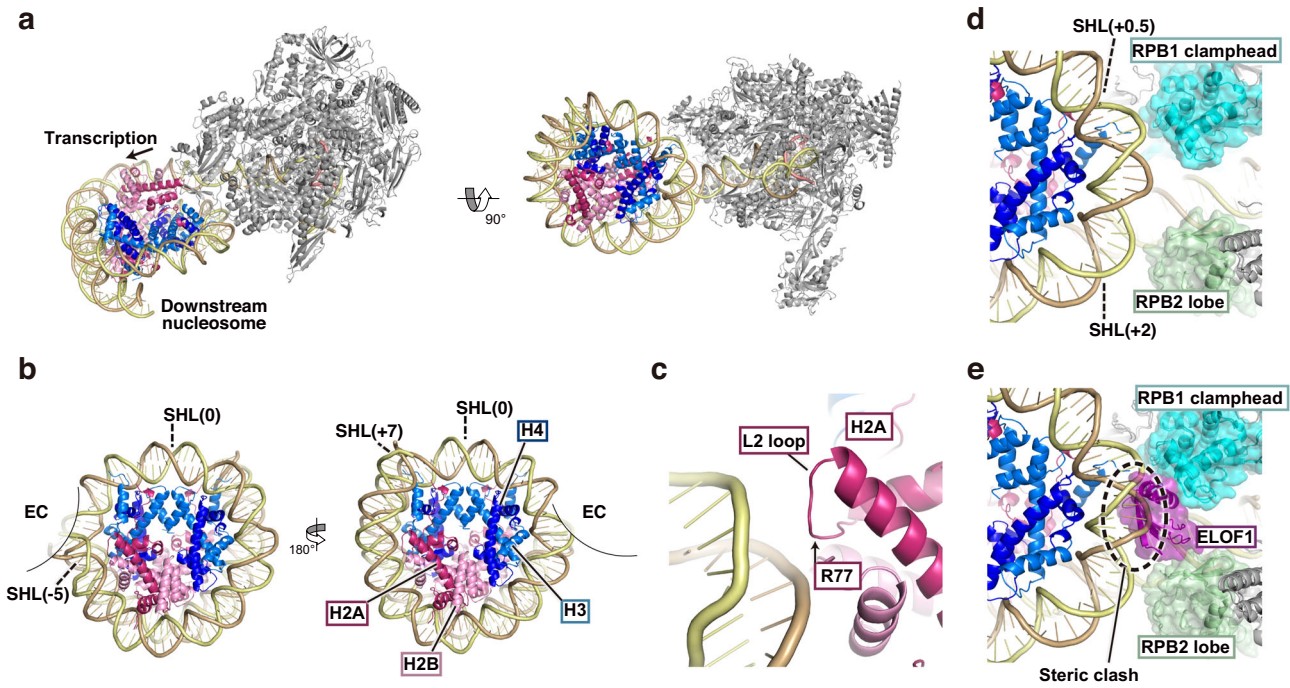

**Fig. 3 | EC-downstream nucleosome complex in human nuclei. a** Structure of the EC-downstream nucleosome. **b** Close-up view of the downstream nucleosome. **c** Close-up view of SHL(−5) on the downstream nucleosome. The position of H2A R77 is shown. **d** The RPB1 clamphead-RPB2 lobe-nucleosomal DNA interaction. The RPB1 clamphead and RPB2 lobe are shown as ribbon models overlaid with transparent surface models. **e** Superimposed view of the EC-downstream nucleosome and the EC-SPT4/5-ELOF1-SPT6. The steric clash between ELOF1 and the nucleosomal DNA is indicated as a dotted circle.

is nearly reassembled on the upstream DNA behind EC (Supplementary Fig. 12a, b)[29]. The nucleosome structures reassembled behind EC were similar between the EC-upstream nucleosome and EC115 structures, in which approximately 120 base pairs of DNA were wrapped around the histone octamer (Supplementary Fig. 12c, d). However, although the upstream nucleosome was directly captured by EC through nucleosomal DNA-EC and exposed H2A-H2B dimer-EC interactions in the EC-upstream nucleosome structure (Fig. 4c, d and Supplementary Fig. 12e), the nucleosome is captured through the interaction between the exposed H2A-H2B dimer and the Spt4-Spt5 KOW1 domain in the previous EC115 complex (Supplementary Fig. 12f). These differences suggest that the RNAPII RPB2 wall, which provides an H2A-H2B contact site, may compensate for the nucleosome reassembly function of the elongation factors.

## Discussion

RNAPII reportedly exists as multiple forms in cells[11,42–44]. To study natural RNAPII structures transcribing genomic DNA in cells, we established the ChIP-CryoEM method, in which the cellular RNAPII complexes with genomic DNA and/or nucleosomes can be isolated from human cells, and determined their structures by cryo-EM. We successfully visualized four structures of the RNAPII ECs with DNA or nucleosomes existing in human nuclei: (i) EC transcribing genomic DNA, (ii) EC transcribing genomic DNA with SPT4/5, ELOF1, and SPT6, (iii) EC paused in the downstream nucleosome, and (iv) EC with the upstream (reassembled) nucleosome. Except for the EC-SPT4/5-ELOF1-SPT6 complex, the elongation factors bound to EC were not visualized, perhaps because they dissociated during sample preparation, especially when the beads bound to RNAPII were washed. It is also possible that the EC without elongation factors may be bona fide transcription

machinery for certain genome regions and/or a temporal state in cells. Further studies will be needed to solve this issue.

It is intriguing that we observed native EC paused at the SHL(-5) position in the downstream nucleosome in human cells (Fig. 3). This is the average structure of the EC-nucleosome complexes transcribing multiple regions of the human genome. Therefore, the DNA and RNA sequences in the complex should be heterogeneous. Despite such heterogeneity, the structure of the EC-nucleosome complex is clear. This fact indicates that the EC pausing at the SHL(-5) position of the nucleosome is not a DNA sequence-specific event, but instead may be a common feature in chromatin transcription. Consistently, previous genome-wide analyses revealed that EC predominantly pauses at the SHL(-5) position of the nucleosome in cells, especially in the first nucleosome (+1) from transcription start sites[24–26]. In the nucleosome, the H2A-H2B dimer located proximal to the incoming EC functions to induce the SHL(-5) pausing by the H2A-DNA interaction (Fig. 3c)[23]. Histone modifications that weaken the histone-DNA interaction around the SHL(-5) position may reduce the SHL(-5) pausing of EC. The chromatin remodeller CHD1, which facilitates nucleosome repositioning, may also be important to reduce this nucleosomal entry barrier. Yeast Chd1 binds nucleosomal histones and DNA around entry/exit sites, and reportedly promotes nucleosome transcription in vitro[45,46]. In contrast, the negative elongation factor NELF increases this nucleosome entry barrier near transcription start sites[47–49]. These factors modulating the nucleosome entry barrier may regulate transcription elongation in chromatin.

Nucleosome reassembly during transcription elongation is proposed to occur by two mechanisms, the elongation factor/histone chaperone-mediated and the template loop pathways[11,50,51]. Consistent with the elongation factor/histone chaperone-mediated pathway,

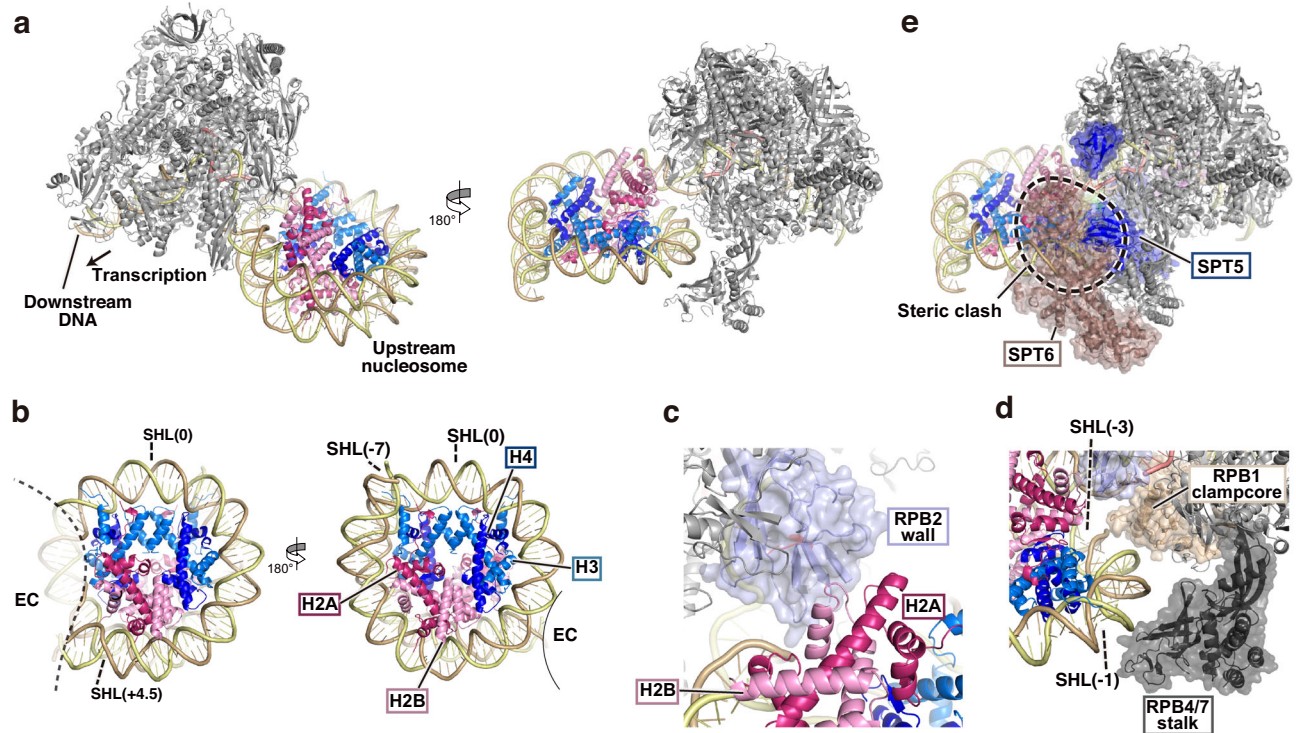

**Fig. 4 | EC-upstream nucleosome complex in human nuclei. a** Structure of the EC-upstream nucleosome. **b** Close-up view of the upstream nucleosome. **c** The RPB2 wall and the nucleosomal H2A-H2B interaction. The wall is shown as ribbon models overlaid with transparent surface models. **d** Close-up view around the RPB4/7 stalk, RPB1 clampcore, and upstream nucleosome. The stalk and clamp core are shown as ribbon models overlaid with transparent surface models.
**e** Superimposed view of the EC-upstream nucleosome and the EC-SPT4/5-ELOF1-SPT6. The steric clash between SPT5, SPT6, and the nucleosomal DNA is indicated by a dotted circle.

snapshot cryo-EM structures during the nucleosome reassembly processes promoted by the yeast EC containing Spt4/5, Elf1, Paf1C, Spt6, and Spn1 were obtained in the presence of the histone chaperone FACT[29]. In the previously reported ECs, the nucleosome reassembly occurs on the upstream EC surface formed by Spt4/5, Paf1C, and Spt6 with the aid of FACT[29]. Conversely, the histone transfer may be promoted by the upstream template DNA looping[30,31,52,53]. In the present EC-nucleosome structure with the upstream nucleosome, the nucleosome is captured by the RPB2 wall, which may promote nucleosome reassembly. Although the dissociation of elongation factors may occur during sample preparation, this structure suggests that the RNAPII RPB2 wall may function in nucleosome reassembly in the elongation factor-independent pathway. A histone chaperone such as FACT may assist to promote this process.

We next determined the EC-SPT4/5-ELOF1-SPT6 complex structure (Figs. 1d and 2). This structure reveals that human ELOF1 functions together with SPT4/5 and SPT6, similar to the yeast counterparts[18]. The overall structure of the EC-SPT4/5-ELOF1-SPT6 complex purified from human nuclei is consistent with those of ECs reconstituted with recombinant human SPT4/5 and SPT6 and yeast Spt4/5, Elf1, and Spt6[10,14,19,29,37-39]. The present EC-SPT4/5-ELOF1-SPT6 complex lacked the elongation factor PAF1C. PAF1C is dynamically associated with and detached from RNAPII, and its localization may be regulated depending on the genomic region and/or environmental factors[36,54]. Therefore, the EC-SPT4/5-ELOF1-SPT6 complex found here may be a unit from which PAF1C could have dissociated during sample preparation and/or in a biological context.

The present ChIP-CryoEM method paves the way for the structural analysis of chromatin-bound complexes. This method allows the acquisition of ~3-4 Å resolution structural information of native target

protein complexes bound to chromatin, and especially, nucleosome-bound complexes. Specific antibodies for histone modifications and variants may be useful to capture native structures of nucleosomes containing these epigenetic factors. Therefore, the ChIP-CryoEM method will reveal novel aspects of chromatin structure and explain the mechanisms of protein functions in native chromatin. In addition, the ChIP-CryoEM method has the potential to identify unexpected novel chromatin complexes containing a target protein by structural analyses. Indeed, we observed larger complexes than the structures of the EC-nucleosome in the cryo-EM images. However, their structures were not obtained because the particle numbers were insufficient, and/or the particles might be too heterogeneous to determine their structures. Note that we crosslinked the immunopurified samples by the GraFix method because samples prepared from crosslinked nuclei tend to aggregate, which may not be appropriate for cryo-EM analysis. Immunopurification before sample crosslinking may induce the dissociation of weakly associating subunits in macromolecular complexes. It is also possible that crosslinkers may block a target epitope for immunopurification, especially for the FLAG epitope peptide containing Lys residues. To reveal the structures of complexes in which subunits are weakly associated or not detected by ChIP-CryoEM, the development of a method for sample preparation from crosslinked cells is awaited.

In this study, we did not visualize the EC-SPT4/5-ELOF1-SPT6-nucleosome complexes. The elongation factors facilitate chromatin transcription with drastic deformation of the nucleosome[12,15,16,19,27-29]. In fact, in vitro nucleosome transcription by RNAPII with elongation factors revealed that most of the nucleosomal DNA region is exposed during the disassembly and reassembly stages[29]. These deformed nucleosomes should be quite sensitive to MNase. Therefore, the

deformed nucleosome in the EC-nucleosome complexes may have been digested by the MNase treatment during the sample preparation. An adaptation of the ChIP-CryoEM method with a specific nuclease such as Tn5, which preferentially attacks certain DNA regions in the nucleosome, could solve this problem[55].

## Methods

### Purification of RNA polymerase II complexes in HeLa cell nuclei

The HeLa cell line expressing hexahistidine-FLAG-tagged RPB3 was established previously[35], and tagged RPB3 expression was confirmed by western blot, using an anti-RPB3 antibody (BETYL, Cat#A303-771A). The HeLa cells were grown in suspension culture using Minimum Essential Medium Eagle with the Joklik modification (Sigma Cat#M0518), at a density of ~$3 - 6 \times 10^5$ cells/ml. The cultured cells were collected by centrifugation at $500 \times g$ for 10 min at 4 °C, washed with cold PBS supplemented with 1 mM $MgCl_2$ and 0.5 mM DTT three times, and resuspended in buffer A (10 mM HEPES-KOH (pH 7.9), 10 mM KCl, 1.5 mM $MgCl_2$, and 0.5 mM DTT). After a 10 min incubation, the cells were centrifuged at $500 \times g$ for 10 min at 4 °C and the supernatant was removed. The resulting swelled cells were resuspended in buffer A and disrupted using a Dounce homogenizer (loose). The resulting cell lysate was centrifuged at $1000 \times g$ for 10 min at 4°C, and the supernatant was removed. The pellet was resuspended in buffer A and centrifuged at $13,600 \times g$ for 20 min at 4 °C. The supernatant was removed, and the pellet was collected as nuclei.

Nuclei of ~$2.0 \times 10^9$ cells were suspended in ~100 ml of ChIP buffer (10 mM Tris-HCl (pH 8.0), 200 mM KCl, 1 mM $CaCl_2$, 0.3% NP-40, 4 mM NaF, 10 mM β-glycerophosphate, and 1x Protease Inhibitor cocktail (EDTA free, Nacalai Tesque)) and loosened by sonication. The resulting sample was treated with micrococcal nuclease (NEB) at 30 °C to digest the chromatin to approximately mono-nucleosome units, and the reaction was stopped by adding 25 mM EDTA. The resulting sample was ultracentrifuged at $68,300 \times g$ for 1 hour at 4 °C, and the supernatant was collected as solubilized chromatin. The chromatin digestion was confirmed by agarose gel or polyacrylamide gel electrophoresis (PAGE) with ethidium bromide staining.

To capture the FLAG-tagged RNAPII, 1,700 µl of the anti-FLAG M2 antibody beads (Sigma), which were pre-blocked with bovine serum albumin, were mixed with the solubilized chromatin and incubated overnight at 4 °C. After the incubation, the beads were collected by centrifugation at $2300 \times g$ for 5 min, 4 °C, and then washed eight times with ~20 CV of wash buffer 1 (10 mM HEPES-NaOH (pH 7.5), 150 mM NaCl, 10 µM $ZnCl_2$, 1 mM DTT, and 0.3% NP-40), followed by three washes with wash buffer 2 (10 mM HEPES-NaOH (pH 7.5), 150 mM NaCl, 0.1% glycerol, and 0.003% NP-40). The RNAPII complexes were eluted with elution buffer (10 mM HEPES-NaOH (pH 7.5), 150 mM NaCl, 0.003% NP-40, 0.1% glycerol, and 0.3 mg/ml FLAG peptide). The elution of the RNAPII was confirmed by SDS-PAGE with Oriole staining (Bio-Rad).

### Identification of the RNAPII-associated proteins by LC-MS/MS analysis

After electrophoresis, the sample lane was divided into 13 sections (5 mm height) and analyzed by LC-MS/MS. The separated gels were cut into pieces and transferred into 1.5 ml tubes. The gel pieces were destained by an incubation with 50 µl of 50% acetonitrile in water and washed with 100 mM ammonium bicarbonate. Acetonitrile (100 µl) was then added to dehydrate the gels, and the supernatant was discarded after several pipetting steps. The gel pieces were reduced by incubation with 50 µl of 10 mM DTT in 100 mM ammonium bicarbonate at 60 °C for 30 min. The supernatant was discarded, and the gels were incubated with 50 µl of 25 mM iodoacetamide in 100 mM ammonium bicarbonate for 60 min at room temperature with shaded lights, for alkylation. The pieces were washed twice in water for 5 min and then twice in 100% acetonitrile for 5 min. The supernatant was

discarded, and the gel pieces were completely dried. The pieces were incubated with 20 ng Trypsin Gold (Promega) in 50 mM ammonium bicarbonate for 12 h, to digest the proteins. The supernatant was transferred to a new tube. Water (40 µl) was added to the gel pieces, and then the supernatant was added to the same tube. Next, 40 µl of 0.1% trifluoroacetic acid (TFA) in 25% acetonitrile was added to the pieces, and the supernatant was added to the same tube. This process was continued in the same manner with 0.1% TFA in 50, 75, and 100% acetonitrile to completely dehydrate the gels. The collected supernatant was completely dried, dissolved in 10 µl of 0.1% TFA, and analyzed by LC-MS/MS. The Nano LC-MS/MS analysis was conducted with an Orbitrap Fusion mass spectrometer equipped with an Ultimate 3000 nano UPLC (Thermo Fisher Scientific). The LC-MS/MS data were analyzed using the Proteome Discover 2.4 software and the *Homo sapiens* amino acid database, including only reviewed proteins obtained from the UniProtKB/Swiss-Prot database site. Spectra were searched with a mass tolerance of 10 ppm in the MS mode and 0.6 Da in the MS/MS mode, allowing up to 2 missed cleavage sites. Cys carbamidomethylation was searched as a fixed modification, whereas Met oxidization and Ser/Thr/Tyr phosphorylation were searched as variable modifications. Peptide confidence in the consensus step was set to at least medium. Confidence thresholds of the protein FDR validator were 0.01 as target FDR (strict) and 0.05 as target FDR (relaxed). One RNAPII sample was analyzed. The raw mass spectrometry data used in this study have been deposited to the proteomeXchange Consortium (PXD052434) via the Japan ProteOme STandard (JPOST) repository[56].

### Western blot of the RNAPII-associated proteins

After the binding reaction with the anti-FLAG antibody beads, the beads were washed with wash buffers containing different concentrations of NaCl (50, 100, and 150 mM). The FLAG-tagged RNAPII was eluted by elution buffers containing different concentrations of NaCl. Equal amounts of the eluted samples were subjected to SDS-PAGE, and proteins were visualized by Oriole staining (Bio-Rad) or western blotting. For the western blots, the proteins were transferred to PVDF membranes (0.22 µm, Amersham, #10600021). The resulting membranes were cut and blocked in Blocking One (Nacalai Tesque #03953-95). The membranes were washed with TBS-T (20 mM Tris-HCl (pH 7.5), 137 mM NaCl, and 0.1% Tween-20) three times and then incubated overnight with anti-FLAG M2 antibody (Sigma # F3165, 1:3000), anti-SPT5 antibody (Proteintech#16511-1-AP, 1:1000), anti-SPT6 antibody (Cell Signaling #15616, 1:2000), anti-PAF1 antibody (Cell Signaling #12883, 1:300), anti-RTF1 antibody (Proteintech #12170-1-AP, 1:1000), anti-MED17 antibody (Proteintech#11505-1-AP, 1:2000), or anti-XPB antibody (Cell Signaling #8746, 1:300) in Can Get Signal Immunoreaction Enhancer Solution 1 (TOYOBO #NKB-101). For detection of the Serine 2phosphorylation, the membrane was incubated with the anti-S2phospho antibody (Merck Millipore #04-1571, 1:2000) for 1 h. The membranes were washed three times with TBS-T and then incubated with the anti-IgG (H + L chain) Mouse pAb-HRP (MBL #330, 1:3000), anti-IgG(H + L chain) rabbit pAb-HRP (MBL #458, 1:3000), or Goat anti-Rat IgG (H + L) antibody HRP (Invitrogen #62-9520, 1:3000) in Can Get Signal Immunoreaction Enhancer Solution 2 (TOYOBO #NKB-101). After three washes with TBS-T, the chemical luminescence signal was detected using ECL Prime Reagents (Cytiva) by an Amersham Imager 680.

The sample eluted under the 150 mM NaCl conditions was subjected to sucrose gradient fractionation without fixation. The eluted RNA polymerase II sample was concentrated to ~1 ml by using an Amicon Ultra 100 K filter (Millipore) and applied to the top of the sucrose gradient. The sucrose gradient was prepared using sucrose buffer (20 mM HEPES-NaOH (pH 7.5), 20 mM NaCl, and 10% sucrose) and sucrose buffer containing 40% sucrose by using a Gradient Master (SKB). The samples were applied to the top of each gradient and centrifuged at 27,000 rpm at 4 °C for 16 h, using a Beckman Colter

SW41Ti rotor. After centrifugation, 800 μL fractions were carefully collected from the top of the gradient solution by pipetting. The fractions were analyzed by non-denaturing PAGE with SYBR Gold staining to detect nucleic acids (DNA and/or RNA). The fractionated samples were subjected to SDS-PAGE, and proteins were visualized by western blotting as described above. The membranes were incubated overnight with anti-FLAG M2 antibody (Sigma # F3165, 1:3000), anti-S2phospho antibody (Merck Millipore #04-1571, 1:2000), anti-SPT5 antibody (Cell Signaling #9033S, 1:1000), anti-SPT6 antibody (Cell Signaling #15616,, 1:2000), anti-PAF1 antibody (Cell Signaling #12883, 1:1000), anti-MED17 antibody (Proteintech#11505-1-AP, 1:2000), or anti-XPB antibody (Cell Signaling #8746, 1:1000). The chemical luminescence signals were detected by an Amersham Imager 680. Source data are provided as a Source Data file.

## Cryo-EM sample preparation

The eluted RNA polymerase II sample was concentrated to ~1 ml by using an Amicon Ultra 100 K filter (Millipore). The concentrated samples were partially purified and stabilized by the gradient-fixation (GraFix) method[57]. The sucrose gradient was prepared using sucrose buffer (20 mM HEPES-NaOH (pH 7.5), 20 mM NaCl, and 10% sucrose) and sucrose buffer containing 40% sucrose and 0.1% glutaraldehyde by using a Gradient Master (SKB). The samples were applied to the top of each gradient and centrifuged at 27,000 rpm at 4 °C for 16 hours, using a Beckman Colter SW41Ti rotor. After centrifugation, 800 μL fractions were carefully collected from the top of the gradient solution by pipetting. The fractions were analyzed by non-denaturing PAGE with SYBR Gold staining to detect nucleic acids (DNA and/or RNA). The fractions containing DNA and/or RNA without aggregation were collected (fractions 4-11, Supplementary Fig. 1d) and dialyzed in dialysis buffer (20 mM HEPES-NaOH (pH 7.5), 20 mM NaCl, 0.2 μM ZnCl$_2$, 0.1 mM TCEP). The resulting sample was concentrated by using an Amicon Ultra 100 K filter (Millipore).

A 2.0 μL portion of each sample was applied to Quantifoil Cu R1.2/1.3 200 mesh grids, which had been glow-discharged for 1 min by a PIB-10 Ion Bombarder (Vacuum Device Inc.). The grids were plunge frozen in liquid ethane using a Vitrobot Mark IV (Thermo Fisher Scientific) at 4 °C and 100% humidity.

## Cryo-EM analysis

The cryo-EM images were collected on a Krios G4 cryo-electron microscope (Thermo Fisher Scientific) equipped with a BioQuantum K3 imaging filter (GATAN) with a slit width of 20 eV, operated at 300 kV with a pixel size of 1.06 Å. In the five data collections, a total of 32,957 movies were recorded using the EPU software (Thermo Fisher Scientific). Each movie was fractionated to 40 frames with a total dose of ~60 e$^-$/Å$^2$ (see Table 1). The following processes were performed using RELION 4[58,59]. Movie frames were aligned and dose-weighted with MotionCor2[60]. The contrast transfer function was estimated by CTFFIND4[61]. Particles were picked automatically with a box size of 120 × 120 pixels and a pixel size of 3.18 Å (3 × binning) by auto-pick, using Topaz[62], and bad particles were removed by 2D classification. After 2D classification, the particles were re-extracted with a box size of 240 × 240 pixels and a pixel size of 2.12 Å (2 × binning). The initial model was prepared using an SGD-based de novo initial 3D model generation. This initial model generation yielded the RNAPII structure (Supplementary Fig. 2c). Then, the initial 3D classification was performed using the obtained RNAPII structure as the reference. In the first 3D classification (Class3D_1), particles were divided into the RNAPII monomer and the other classes. The RNAPII monomer classes were further classified (Class3D_2) and divided into the RNAPII elongation complex (EC), DNA-free RNAPII, and DNA-free RNAPII without stalk classes. The EC classes were further classified (Class3D_3) and divided into the EC-DSIF-SPT6-ELOF1, EC, EC with the upstream nucleosome, and EC with the downstream nucleosome classes. The

other classes in Class3D_1 were classified (Class3D_4) and divided into the EC-nucleosome and the other classes. The EC-nucleosome classes were further classified (Class3D_5) and divided into the EC-upstream nucleosome and other classes. The other classes in Class3D_5 were further classified (Class3D_6), and the EC-downstream nucleosome class was obtained.

For the EC structure, the selected particles were subjected to re-extraction with the original pixel size (1.06 Å/pixel), CTF refinement, Bayesian polishing, refinement, and postprocessing. The resolution of the final map of the RNAPII is 2.7 Å.

For the EC-DSIF-ELOF1-SPT6 structure, the selected particles were subjected to re-extraction with the original pixel size (1.06 Å/pixel), CTF refinement, Bayesian polishing, and refinement. The RPB4/7 stalk-SPT6 part was subtracted and classified (Class3D_7). The resolved classes of the SPT6 were selected, refined, and postprocessed. The resolution of the final map of the RPB4/7 stalk-SPT6 is 3.5 Å. The particles were then reverted to the original images, refined, and postprocessed. The resolution of the final map of the EC-DSIF-SPT6-ELOF1 complex is 3.1 Å.

For the EC-downstream nucleosome, the relevant classes in Class3D_3 and Class3D_6 were combined. The combined particles were subjected to re-extraction with the original pixel size (1.06 Å/pixel), CTF refinement, Bayesian polishing, and refinement. The nucleosome part was subtracted and classified (Class3D_8). The resolved nucleosome class was selected, refined, and postprocessed. The resolution of the final map of the downstream nucleosome is 4.3 Å. The particles were then reverted to the original images, refined, and postprocessed. The resolution of the final map of the EC-downstream nucleosome is 4.1 Å. The RNAPII part of the EC-downstream nucleosome was subtracted, refined, and postprocessed. The resolution of the final map of the RNAPII is 3.7 Å.

For the EC-upstream nucleosome, the EC-upstream nucleosome classes in Class3D_3 and Class3D_6 were combined. The combined particles were subjected to re-extraction with the original pixel size (1.06 Å/pixel), CTF refinement, Bayesian polishing, and refinement. Then, the nucleosome part was subtracted and classified (Class3D_9). The resolved nucleosome class was selected, refined, and postprocessed. The resolution of the final map of the upstream nucleosome is 3.6 Å. The particles were then reverted to the original images, refined, and postprocessed. The resolution of the final map of the overall RNAPII-upstream nucleosome is 3.6 Å. The EC part of the EC-upstream nucleosome was subtracted, refined, and postprocessed. The resolution of the final map of the RNAPII is 3.3 Å.

The resolutions of all final maps were estimated by the gold standard Fourier shell correlation (FSC) = 0.143[63]. The local resolutions were estimated using RELION.

## Model building

The atomic model of EC was built based on the cryo-EM structure of the *Bos taurus* EC (PDB ID: 5FLM)[64]. The model of the *Bos taurus* EC was rigid-body fitted into the cryo-EM density map of RNAPII by UCSF Chimera[65]. The amino acid residues of the *Bos taurus* RNAPII were adjusted to those of human RNAPII. The resulting model was refined using phenix.real_space_refine[66,67], and edited manually using Coot[68] and ISOLDE[69] in UCSF ChimeraX[70].

For building the EC-nucleosome complex models, the starting models were generated based on the previous cryo-EM structures of the *Komagataella pastoris* EC-nucleosome complexes (PDB ID: 6A5P)[23] for EC-downstream nucleosome, PDB ID: 7XSZ for EC-upstream nucleosome[29]. The *K. pastoris* RNAPII parts and the nucleosomal histone parts were replaced by the human RNAPII structure modeled above and the histones of the high-resolution nucleosome structure (PDB ID: 7VZ4), respectively. The resulting EC parts and nucleosome parts were independently fitted by Chimera[65] and refined against the focused refined maps using phenix.real_space_refine[66,67]. Subsequent manual adjustments were performed with Coot[68] and ISOLDE[69] within

ChimeraX[70] to improve the geometric quality of the models and the model-to-map fitting. The resulting partial models were then assembled into the overall structures.

The atomic model of the EC-SPT4/5-ELOF1-SPT6 complex was built based on the cryo-EM structure of the *Sus scrofa* EC-DSIF-PAF1C-SPT6 complex (PDB ID: 6GMH)[14]. The ELOF1 model was built based on the AlphaFold2 model[71]. The PAF1C, DSIF, and SPT6 models were removed from the *Sus scrofa* EC-DSIF-PAF1C-SPT6 structure, and the resulting RNAPII model and ELOF1 were rigid-body fitted into the cryo-EM density map of the EC-SPT4/5-ELOF1-SPT6 complex by UCSF Chimera[65]. The amino acid residues of the *Sus scrofa* RNAPII were adjusted to those of human RNAPII. The SPT4 and SPT5 NGN and KOW domains were separately fitted with the EC-SPT4/5-ELOF1-SPT6 map. The resulting model was refined using phenix.real_space_refine[66,67], and edited manually using Coot[68]. The model of the SPT6 core was built based on the EC-CSB-CSA-DDB1-UVSSA-PAF1C-SPT6 structure (PDB ID: 7OOP)[72] and fitted with the RPB4/7 stalk into the RPB4/7 stalk-SPT6 map. The RPB4/7-SPT6 part was refined using phenix.real_space_refine[66,67], and edited manually using Coot[68]. The resulting RPB4/7-SPT6 model was combined with the EC-ELOF1-DSIF model by fitting into the EC-SPT4/5-ELOF1-SPT6 map, and edited manually using Coot[68].

All of the final structures were evaluated using MolProbity (Table 2)[73]. All structural figures were produced with PyMOL (Schrödinger; http://www.pymol.org) and UCSF Chimera[65].

## The use of large language Models
GPT-4o and ChatGPT3.5 were used for grammatical correction of the text and supplied the basis of the Python programs. No original sentence was produced by AI. The programs were used to process PDB files. We generated the following four programs to process the PDB files. (1) A program that extracts and returns the residues that are clash pairs in the output of the MolProbity software, in a form suitable for ISOLDE. (2) A program that extracts and returns the residues whose coordinates are different between two PDB files, as a list. (3) A program that erases "TER" on the line following "HETATM" and adds "END" on the line below the last "TER". (4) A program that extracts and returns only CABLAM outliers, using CABLAM results as input. Grammarly (ver. 1.56.1.0) was also used for grammatical correction.

## Reporting summary
Further information on research design is available in the Nature Portfolio Reporting Summary linked to this article.

## Data availability
The cryo-EM reconstructions and atomic models of the RNAPII complexes generated in this study have been deposited in the Electron Microscopy Data Bank and the Protein Data Bank (PDB) under the accession codes: EC: PDB ID 8XSO and EMDB entry ID EMD-38624; EC-SPT4/5-ELOF1-SPT6: PDB ID 8XRM and EMDB entry ID EMD-38607; EC-downstream nucleosome: PDB ID 8XVS and EMDB entry ID EMD-38717; EC-upstream nucleosome: PDB ID 8XRJ and EMDB entry ID EMD-38604.The raw mass spectrometry data generated in this study have been deposited to the proteomeXchange Consortium (PXD052434) [https://proteomecentral.proteomexchange.org/cgi/GetDataset?ID=PXD052434] via the Japan ProteOme STandard (JPOST) repository under the accession code: JPST003120. Source Data are provided with this paper. Previously published PDB codes used in this study: 7XN7; 9EH2; 6TED; 6A5P; 7XSZ. Source data are provided in this paper.

## Code availability
This paper does not report original codes.

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

## Acknowledgements

We thank Y. Iikura, Y. Takeda, T. Ito (The University of Tokyo), M. Kato, and M. Ito (Kyushu University) for their assistance. We thank the MEXT Promotion of Development of a Joint Usage / Research System Project:Pan-Omics DDRIC, MRCI for High Depth Omics, CURE:JPMXP1323015486 for MIB and RIIT in Kyushu University. This work was supported in part by JSPS KAKENHI Grant Numbers JP20H03201 [to T.K.], JP20H05690 [to T.K.], JP22K15033 [to T.K.], JP23K17392 [to T.K.], JP24H00062 [to T.K.], JP22KJ0858 [to S.H.], JP22K06098 [to Y.T.], JP23H05475 [to H.K.], JP24H02328 [to H.K.], JP22J20655 [to T.F.], JP22H04696 [to K.M.], JP23H04288 [to K.M.], JP21H05292 [to A.H.], JP23H02394 [to A.H.], JP23H00372 [to Y.O.], JP24H02323 [to Y.O.], JP22H04676 [to Y.O.], JP22K19275 [to Y.O.], and JP20H03182 [to Y.Y.], JST ERATO Grant Number JPMJER1901 [to H.K.], JST CREST Grant Number JPMJCR24T3 [to H.K.], JPMJCR24Q1 [to K.M], JPMJCR23N3 [to K.M], JST PRSTO Grant Number JPMJPR2026 [to K.M.], AMED-CREST Grant Number JP23gm1810008 [to A.H.], and AMED BINDS Grant Numbers JP25ama121009 [to H.K.], JP25ama121002 [to Y.T], and JP23ama121017j0001 [to Y.O.].

## Author contributions

Conceptualization: T.K. and H.K. Methodology: T.K., J.K., K.Y., S.H., T.F., K.M., A.H., Y.O., L.N., and Y.T. Resources: T.K., J.K., K.Y., S.H., L.N., M.O., Y.Y., Y.T., and H.K. Investigation: T.K., J.K., K.Y., S.H., L.N., T.F., K.M., A.H., Y.O., Y.T., and H.K. Visualization: T.K., S.H., Y.T., and H.K. Funding acquisition: T.K., S.H., Y.Y., Y.O., Y.T., and H.K. Project administration: T.K. and H.K. Supervision: H.K. Writing – original draft: T.K. and H.K. Writing – review & editing: T.K., Y.O., Y.T., and H.K.

## Competing interests

The authors declare no competing interests.
