## [Peer Review file · Nature Communications]

Multiple structures of RNA polymerase II isolated from human nuclei by ChIP-CryoEM analysis

Corresponding Author: Professor Hitoshi Kurumizaka

Version 0:

Reviewer comments:

Reviewer #1

(Remarks to the Author)

Kujirai and colleagues present multiple structure of RNAPII isolated through chromatin immunopurification. The structures of complexes with or without elongation factors and associated with nucleosomes in various stages of nucleosome uncoiling and recoiling confirm previous reconstitution work and provide interesting elements of novelty. For example they observe that pausing of DNA uncoiling from histones occurs at SHL (-5) in a sequence independent manner, which is in agreement with genome-wide analysis of RNAP II pausing. Technically, the work is very nicely done and the conceptual novelty provided is appropriate for publication in Nature Communications. A list of minor points to be addressed is found below.

Minor issues

1. Page 3 line 18. In citing the beads-on-a-string model the authors should consider citing more modern work reporting structural evidence from nuclei imaged in situ. For example: <https://doi.org/10.1038/s41467-023-42072-1>
2. End of the introduction: Can the authors justify why HeLa cells have been chosen in particular?
3. Regarding the fragility of RNAP II-chromatin complexes, the authors discuss potential shortcomings on their approach both in the results and in the discussion. They explain that running GraFix after chromatin digestion immuno-affinity purification could mean that nucleoprotein complexes come apart, but that crosslinking chromatin causes unmanageable aggregation. Have the authors considered crosslinking after immunopurification and before running a gradient?
4. Page 5/6 line 45-4. I feel that the observation of a structure with RNAPII paused at SHL(-5) and its correlation with genome wide studies is one of the more compelling findings in this study. It features in the abstract but it seems not to occupy central enough a spot in the main text. The authors could consider mentioning the genomic study in the introduction and highlighting the finding more in the results and discussion. This would help the reader better appreciate the conceptual advance brought by determining RNAPII chromatin structures from cells.
5. Page 6 lines 5-14. I feel the authors should discuss the possibility that direct nucleosome RNAPII contacts are established after elongation factors fall off. If there are technical reasons why they don't think this is a possibility, they should address the issue.
6. Discussion: the authors claim that they established a ChIP and cryo-EM method to solve endogenous structures. How does this approach differ from similar methods used to image chromatin bound complexes purified from cells?

Language

A few sentences should be revised.

Abstract:

- Line 10: corresponding disassembly process. Please correct.
- Line 14 "during nucleosome reassembly process" should be "during nucleosome reassembly" or "during the nucleosome reassembly process".
- Line 15 "transcribing human genome" should be "the human genome".

Main text:

- Page 3 line 12: “two each of these histones” should be revised.
- Line 13 “in which” should be “around which”.
- Page 4 line 4. Associated and not associating.
- Page 5 line 6. I suggest “also represent an average of heterogeneous sequences”.

Reviewer #2

(Remarks to the Author)

Kujirai et al. report cryo-EM structures of RNA polymerase II (RNAPII) transcription elongation complexes (ECs) obtained by a novel immunoprecipitation method from nuclear extracts of live human tissue culture cells. This approach is important because it can give insight into the states of ECs and associated factors in living cells. As the first well-executed report of this approach, the manuscript is appropriate for high-visibility publication. ECs collected from living cells are expected to be in a variety of states because the EC associates with different factors as it transitions from transcript initiation to elongation through nucleosomes, mRNA splice sites, and polyA-addition sites, and then finally to a termination state after the mRNA is cleaved at the polyA site. The authors are able to identify four types of complexes by computational particle sorting of the heterogeneous EC mixture recovered by immunoprecipitation: (1) an EC containing only RNAPII, DNA, and RNA; (2) an EC containing elongation factors Spt4, Spt5, Spt6, and Elof; (3) an EC that has encountered a nucleosome and is halted at so-called nucleosome position SHL(-5); and (4) an EC that is exiting a nucleosome and is halted at so-called position SHL(+5). The major finding is simply that these complexes can be recovered by immunoprecipitation and analyzed by cryo-EM. This is an important advance because it validates decades of RNA polymerase structural analyses that have relied on purified proteins and reconstituted complexes. Although additional insights are somewhat limited, the overall story merits publication. That said, major revision is required. The authors suggest that the absence of elongation factors in all but the second complex indicates that ECs in vivo bind and release elongation factors as a function of template position or relationship to a nucleosome. Although possible, this conclusion is not supported by their findings because it is equally likely the factors dissociated during the course of sample preparation. A more balanced interpretation of their results will be required to merit peer-reviewed publication.

Major points.

1. As noted in the summary, the conclusion that some ECs, in particular those that are entering and departing a nucleosome, lack elongation factors is not supported by the authors results. The isolation of complexes involved multiple wash steps before crosslinking is applied to stabilize the complexes. There's nothing wrong with this approach, but it precludes conclusions about whether factors are present on the complexes at the point of nuclei isolation. The authors should discuss the potential impact of the isolation procedure in a more balanced fashion that makes explicit the inability to draw conclusions about the native state of the complexes.
2. The authors label their EC lacking elongation factors an ‘RNAPII’ complex rather than an elongation complex (EC) and then use the EC nomenclature to refer to the EC plus elongation factors. This choice directly contradicts over two decades of precedent in the scientific literature and the transcription field and will confuse readers. An EC is a complex of RNAPII, RNA, and DNA engaged in transcript elongation. The authors ‘RNAPII’ complex is an EC. RNAPII might be present in the sample, but the authors do not comment on whether or not they see evidence for RNA–DNA-free RNAPII (they might add this to the discussion). The authors should adopt a different nomenclature consistent with precedent and refer to all ECs as ECs. For example, they could use EC- and EC+ to indicate ECs with and without elongation factors, respectively (or any number of other alternatives).
3. The authors state that their complexes are pretranslocated. Their cryo-EM maps do not support this simplistic statement and rather suggest the complexes are mixtures of pre-translocated and post-translocated ECs as might be expected from a heterogeneous population of ECs isolated from cells. The cryo-EM signal for the modeled 3' RNA nucleotide in the pretranslocated register is significantly weaker than the signal for the penultimate RNA nucleotide. This is a clear indication that some or even many of the ECs lack an RNA nucleotide in the product subsite of the active site and are post-translocated. The interpretation should be modified to accurately reflect the data by stating the complexes are mixtures of pre- and post-translocated ECs, with the fraction of each difficult to assess.
4. The authors rely on an antibody-targeted adapter insertion method they call ChIL-seq to map RNAPII and elongation factor association with DNA in HeLa cells. The authors argue that differences in the patterns for RNAPII ChIL peaks and Spt5–6 peaks indicates that RNAPII is sometimes localized to gene bodies without associated Spt5 and Spt6. These data and analyses are wholly inadequate to support such a conclusion. It's unclear how the histogram of peak distributions across a set of genes (Fig. 1e) even supports the conclusion since it doesn't directly depict the extent of peak co-localization. Beyond that, no statistics are offered to establish that the patterns differ. Genome-scale analyses without robust statistics are largely meaningless. In particular, triplicate datasets would be required and analyses provided showing the pattern differ more between RNAPII and Spt5–6 than between the replicates to statistically significant extent. Absent such analyses and a better illustration of the patterns of peaks observed, the results do not support a conclusion that RNAPII is bound in the body of genes without elongation factors. The single-molecule measurements of Spt5 association and disassociation cited by the authors do not address occupancy of ECs by Spt5 in vivo. This part of the manuscript should be improved or removed.
5. The authors mention that previous genome-scale analyses have shown RNAPII paused at the -5 nucleosome position in vivo. Do the authors' Spt5–6 data (or other published Spt5–6 data) suggest Spt5–6 localize to the -5 nucleosome position or is it depleted from this location? The authors argue their structure of ECs lacking Spt5–6 at this location indicate that the

nucleosome-paused EC lacks Spt5–6. The genome-scale data should be analyzed to directly test if this is true in vivo and the result reported with the authors findings.

6. Some features modeled into the provided pdb files do not appear to be supported by the cryo-EM maps. For example, the portion of the trigger loop near the rim helices and the RNA upstream of RNAPII interacting with Spt6, which are modeled in the EC plus elongation factor structure, do not appear to be supported by corresponding signal in the cryo-EM map. The authors should carefully curate the models for each of the structures they are reporting and remove from them any segments of polypeptide not supported by the cryo-EM data. Additionally, in regions of the models where resolution is clearly insufficient to support modeling of amino acid–side chains (e.g., Spt6 in the EC plus factors model), the authors should limit the model to the Ca backbone.

Minor points

1. The first paragraph of the results cites refs. 1–3 after each sentence. Citing once after the topic sentence is sufficient.
2. line 131 and throughout manuscript. Results should be reported in past tense. "...all twelve protein subunits *were* observed..." (not "are observed").
3. line 218. The authors should explicitly state they are comparing structures of a yeast EC to the human EC results and give the citation to the yeast results at the end of the sentence.
4. Figs. 3b and 4b. Why do the arrows point in opposite directions in the 180° rotated views? This difference will confuse readers trying to relate the two images to each other.

Reviewer #3

(Remarks to the Author)

Kujirai et al., present their findings using an innovative method to prepare RNA Polymerase II (RNAPII) samples for cryo-EM analysis directly from HeLa cells: ChIP-Cryo-EM. Their approach takes advantage of the ability to directly isolate FLAG tagged RNAPII from HeLa cells, followed by a series of washes, sample separation and crosslinking via GraFix, and analysis using single-particle cryo-EM. This methodology successfully identified several RNAPII complexes actively engaged in transcription elongation. The authors describe two RNAPII elongation complexes, one in the absence and one in the presence of elongation factors. Additionally, they report two transcription elongation complexes lacking transcription factors but featuring a nucleosome either upstream or downstream of RNAPII. They suggest that these RNAPII-nucleosome conformations resemble the chromatin disassembly and reassembly processes during transcription.

This manuscript offers a novel approach to capturing transcription complexes that complements traditional in vitro reconstitution methods. Their technique, as described, serves as a proof of concept. However, some key controls are missing, and a few of the claims, particularly the role of an RNAPII elongation complex without elongation factors as a key intermediate of gene-body transcription, are not fully supported by the presented data. Specific comments the authors should address prior to publication are provided below.

Major Comments:

1. ChIL-seq Experiment: The authors use this assay to support their claim that RNAPII is actively transcribing coding sequences without DSIF and SPT6. Several issues complicate this interpretation, and the methodology is described in a way that makes it difficult to follow:
 - o It appears that all three antibodies (anti-FLAG for RNAPII, -SPT5, and -SPT6) were added simultaneously. The varying affinity of each antibody could impact the efficiency of capturing all three proteins when used together. No catalog numbers are provided for the antibodies, making it impossible to evaluate their suitability for this experiment.
 - o The presence of RNAPII elongation complexes is inferred by the colocalization of FLAG, SPT5, and SPT6 ChIL-seq peaks. If FLAG peaks do not overlap with the others, the authors consider this an RNAPII elongation complex without elongation factors. However, RNAPII elongation complexes could lack DSIF or SPT6, and instead include factors involved in transcription coupled repair, the super elongation complex, elongin, NELF, or RNA processing complexes. Additionally, elongation factors like SPT5 can exhibit flexibility without disrupting the entire complex (see further comments on this in point 3 below). Such flexibility could reduce the efficiency of the secondary antibody to insert the transposable sequence near chromatin, especially in nucleosome-rich regions, complicating the interpretation of the results. The authors should consider alternative explanations for the absence of peaks.
 - o The secondary antibody's transposon insertion efficiency and library preparations may introduce biases in peak identification. This issue is not addressed in the experimental design, and these biases could obscure the conclusion that RNAPII alone is transcribing. The authors should discuss these potential biases.
 - o How do these observations compare with ChIP-seq experiments? Previous ChIP-seq data show continuous SPT5 and SPT6 occupancy throughout gene bodies, and, as far as I am aware, there is no evidence of depletion of these factors in the vicinity of nucleosomes.
 - o Finally, the claim that a substantial portion of RNAPII elongation complexes would transcribe through chromatin in the absence of any elongation factors conflicts with numerous studies. It is known that RNAPII elongation complexes are slow and inefficiently overcome the nucleosome barrier without elongation factors in vitro and in cells. The assertion that RNAPII elongation complexes lacking elongation factors represents a physiological state requires more substantial evidence and would be a major deviation from our current understanding of transcription elongation in mammals.

2. ChIP-Cryo-EM:

o Although the authors mention sample fractionation by GraFix, it is unclear whether each fraction was individually frozen on grids. If so, it would be important to note whether the fractions contained different complexes. For example, if the large complexes the authors mentioned at the end of the discussion are enriched in certain fractions, they would be omitted from the data collection. This could include complexes representing initiation complexes, complexes bound to PAF1c, or RNA processing machineries. It appears that the complexes that are presented here would have a molecular weight < 1 MDa, suggesting that larger complexes are lost during the GraFix purification.

o Including a denaturing gel of non-crosslinked GraFix fractions (running a second sample through the sucrose gradient without glutaraldehyde) would help determine the subunit composition of the samples. Additionally, mass spectrometry of each GraFix fraction would clarify the proportion of additional factors associated with RNAPII.

o The authors mention the potential loss of factors during the washing steps, but no controls are provided to demonstrate that these factors are indeed lost in the washes. Additionally, phosphatase inhibitors only appear to be included in the initial wash and not included in later incubation steps. It is known that RNAPII phosphorylation state can greatly affect the stability of the elongation factors that associate with the RNAPII elongation complex. The authors use 150 mM NaCl in their early wash steps, and this salt concentration would cause more weakly associated elongation factors like RTF1 to disassociate from RNAPII. It would be worth mentioning if the authors have tried less stringent washing procedures and suggest this as an additional parameter that can be manipulated in future iterations of this experiment.

3. The authors provide extensive explanations for the interactions between DSIF and ELOF1 and SPT6 and DSIF. In these discussions, they do not mention the KOW5 domain of DSIF. This is the most stably bound portion of DSIF on RNAPII and appears to be present in the provided models. The authors state the DSIF is not present in the nucleosome structures. Have they carefully examined their maps for density corresponding to SPT5 KOW5? The reviewer acknowledges that the other observed DSIF domains would clash with the position of the upstream nucleosome, and these domains would be flexibly tethered to RNAPII by the KOW5 domain.

4. The authors need to adjust their terminology for the complexes they are studying to be consistent with the literature. All complexes the authors investigate structurally are transcription elongation complexes. The description of elongation complexes lacking elongation factors should not be described as “transcribing RNAPII” or “RNAPII alone”. To make it clear, they should call this complex an RNAPII elongation complex. For their other elongation complex with factors, it would be best to call this a RNAPII-ELOF1-DSIF-SPT6 elongation complex. Similarly, use of @DS or @US to describe the location of the nucleosome relative to RNAPII is not intuitive. Here it would be best to describe this as an RNAPII elongation complex associated with an upstream nucleosome (or downstream nucleosome). The use of abbreviations can make it difficult for others in the field and outside of the field to understand the exact nature of the complexes are being studied.

5. Line 195-196- The authors state that the RNAPII upstream nucleosome complex contains “tightly rewrapped” DNA around the nucleosome. They have no evidence to indicate that this nucleosome was actually transcribed in their experiments. It would be best to state that the “DNA is tightly wrapped” to avoid overinterpretation of their data.

6. Line 73- the authors state “before the center of the nucleosomal DNA”. They should call this the nucleosomal dyad.

7. Line 45- The first sentence of the introduction is misleading. RNAPII transcribes non-coding and protein coding regions of the genome, but other RNAPs also can perform these functions. They should rephrase to emphasize the RNAPII primarily transcribes protein coding genes and some non-coding gene regions.

8. Line 47- The authors state “During gene expression, the storage RNAPII molecule”. I am not sure what the “storage RNAPII molecule” is. Do they mean RNAPII dimers or RNAPII bound to RPAP2? To avoid confusion, I would simplify this sentence to state RNAPII is assembled on promoters.

9. The paragraphs starting at lines 257 and 269 respectively, could be swapped to more accurately follow the flow of the presented text.

10. The authors state that they do not observe PAF1c bound to RNAPII in their cryo-EM data. Have they tried to further classify their SPT6 particles? PAF1c is difficult to initially observe in cryo-EM and they may be model biased in their classification approaches. They have 130k particles for the SPT6 EC, and there is a possibility that a fraction of these particles have weak PAF1c density. Also, have the authors looked for SPT6 tSH2 density in their maps using focused classification approaches?

Minor Comments

- Introduction (p. 3, lines 29–38): The description of nucleosome transcription by RNAPII is unclear. It is particularly difficult to distinguish which findings are specific to yeast and which apply to metazoans. The authors should also include reference 50 in their introduction for prior structural work.

- Introduction (p. 3, line 39 onward): The authors describe the complexity of factors involved in chromatin transcription. However, their findings suggest the opposite, that the RNAPII elongation complex is present near nucleosomes without elongation factors, which conflicts with the literature they cite. The authors should address this apparent contradiction, possibly in the discussion section.

- Figure 1D and 1E: The provided graphics are difficult to interpret. It is unclear whether the counts are normalized against the number of peaks or another criterion. If there are more peaks classified as RNAPII alone, it would be expected to see more counts along the coding regions. The presentation of the data is unclear. Adding example gene tracks and correlating the data with nucleosome positioning from previously published MNase-seq could provide better context for interpreting the peaks.

- Discussion (p. 7, lines 26–36): The authors cite Rosen G.A. et al (reference 40). to support their argument but do not address conflicting findings that demonstrate the essential roles of SPT5, SPT6, and PAF1c in transcription elongation. How can elongation factors dissociate without being outcompeted by other binding factors in a cellular context? The only referenced previous works was done with yeast components for a single molecule analysis. Is there any prior evidence of such dynamic binding of DSIF for the human RNAPII?

- Figure 2a: What do the dotted lines represent? Are these the regions that the authors observe in the structure? If so, KOW6-7 has not been visualized in a transcription EC to date (and is included in this box) and the SPT6 tSH2 has been visualized in multiple structures (and is excluded from this box).

- 2D class images: Scale bar is missing. (Fig S2b)
- Supplementary figure 8: Both of these active sites represent elongation complexes. (related to point 4)

Version 1:

Reviewer comments:

Reviewer #1

(Remarks to the Author)

The authors addressed all the points I raised. I feel this manuscript is now ready for publication.

Reviewer #2

(Remarks to the Author)

The authors have substantially revised the manuscript to address the concerns of the reviews. Importantly, the CHIP-seq data, which had been over-interpreted and did not support the conclusions drawn, is removed from the revised manuscript. Further, the statements about ECs lacking elongation factors have been revised to acknowledge that the factors may well have dissociated during isolation of complexes. A number of other revisions have improved the interpretation of the data significantly. As revised, the manuscript now reports an important advance in the elucidation of RNAPII structures from cells. The revised manuscript is largely suitable for publication, but there is still one issue of nomenclature that the authors should fix. Reviewers 2 and 3 both asked that they authors not refer to elongation complexes as RNAPII but as elongation complexes. This distinction is important because RNAPII can exist in many forms, including not bound in active transcription complexes but the authors report exclusively on elongating transcription complexes. References to RNAPII may confuse readers by implying RNA and DNA may not be present in the complexes. Although acknowledging the issue and making some revisions, the confusion still persists in the revised manuscript. This may simply be a matter of translation to English. I think using the EC abbreviation for elongating or elongation complex that was established in the first report of RNAPII EC structures (Gnatt et al., 2001. Science 292:1876) would make the manuscript easier for a general audience to follow. To illustrate this point, I suggest revising the paragraph in the results starting at line 138 to read as follows. This nomenclature could then be used thereafter in the rest of the manuscript.

"We first obtained two distinct native RNAPII elongation complex (EC) structures, EC with RNAPII alone and EC with RNAPII complexed with SPT4/5, ELOF1, and SPT6 at 2.7 Å and 3.1 Å resolutions, respectively (Fig. 1c, d). These ECs were actively transcribing genomic DNA or stalled on certain genomic regions in human cells. In both EC structures, all twelve RNAPII subunits were observed (Fig. 1c, d). Genomic DNA and nascent RNA fragments were clearly visualized around the RNAPII catalytic center, suggesting that they were actively transcribing forms of human RNAPII ECs (Supplementary Fig. 8a,d). Since these are average structures of ECs at various genomic DNA loci, the cryo-EM maps of the DNA and RNA bases represent an average of heterogeneous sequences. Further, these ECs may be mixtures of pre-translocated and post-translocated forms because the densities of the RNA base at the +1 position are relatively weak (Supplementary Fig. 8b,c,e,f). ECs without elongation factors may be observed as a consequence of elongation factor dissociation during sample preparation for the cryo-EM analysis. Alternatively, ECs without elongation factors may function in genome transcription at certain loci. Further studies are required to solve this issue."

Reviewer #3

(Remarks to the Author)

The authors have mostly addressed my previous concerns. Here are a few additional points that should be addressed prior to publication.

1- Lines 166-168 and line 299-300— It is odd the the authors compare their structure to the reconstituted yeast complex when there is a reconstituted mammalian complex, which would more closely resemble what they are studying here. The authors should minimally cite DOI: 10.1038/s41586-018-0440-4

2- Paragraph starting at line 238- The authors again compare their human structures to a *K. pastoris* structure. There are now multiple structures of upstream NCPs on the mammalian EC, and these would be a more appropriate for comparison (can be added in addition to the *K. pastoris* data). See DOIs: 10.1126/science.adn6319 and 10.1101/2024.12.13.628244

3- Supplementary Figure 8-

Panel G is cited, and there is no panel G. It appears that panel D was omitted from the figure legend.

Version 2:

Reviewer comments:

Reviewer #2

(Remarks to the Author)

The authors have satisfactorily revised the manuscript and it is ready for publication.

Reviewer #3

(Remarks to the Author)

The authors have addressed my comments.

REVIEWER COMMENTS

Reviewer #1 (Remarks to the Author):

Kujirai and colleagues present multiple structure of RNAPII isolated through chromatin immunopurification. The structures of complexes with or without elongation factors and associated with nucleosomes in various stages of nucleosome uncoiling and recoiling confirm previous reconstitution work and provide interesting elements of novelty. For example they observe that pausing of DNA uncoiling from histones occurs at SHL (-5) in a sequence independent manner, which is in agreement with genome-wide analysis of RNAP II pausing. Technically, the work is very nicely done and the conceptual novelty provided is appropriate for publication in Nature Communications. A list of minor points to be addressed is found below.

Minor issues

1. Page 3 line 18. In citing the beads-on-a-string model the authors should consider citing more modern work reporting structural evidence from nuclei imaged in situ. For example: <https://doi.org/10.1038/s41467-023-42072-1>

Reply)

We cited more modern work, including this reference.

2. End of the introduction: Can the authors justify why HeLa cells have been chosen in particular?

Reply)

We added a sentence to explain why we used HeLa cells for the establishment of this ChIP-CryoEM method (p.4, l.20-22).

3. Regarding the fragility of RNAP II-chromatin complexes, the authors discuss potential shortcomings on their approach both in the results and in the discussion. They explain that running GraFix after chromatin digestion immuno-affinity purification could mean that nucleoprotein complexes come apart, but that crosslinking chromatin causes unmanageable aggregation. Have the authors considered crosslinking after immunopurification and before running a gradient?

Reply)

We suspected that most of the proteins bound to RNAPII may dissociate during the bead washing step, before running GraFix. In addition, the sample just after the immunopurification contains a large amount of FLAG peptide, which may inhibit efficient crosslinking by glutaraldehyde. Therefore, we did not test the crosslinking after immunopurification and before running a gradient.

4. Page 5/6 line 45-4. I feel that the observation of a structure with RNAPII paused at SHL(-5) and its correlation with genome wide studies is one of the more compelling

findings in this study. It features in the abstract but it seems not to occupy central enough a spot in the main text. The authors could consider mentioning the genomic study in the introduction and highlighting the finding more in the results and discussion. This would help the reader better appreciate the conceptual advance brought by determining RNAPII chromatin structures from cells.

Reply)

Thank you very much for this comment. In the revised manuscript, we rewrote the Results section “The structure of RNAPII paused in the downstream nucleosome in human nuclei” and introduced the previous genomics results in the Introduction, according to this reviewer’s comment. We also added a new Discussion paragraph and discussed the conceptual advancements achieved by determining RNAPII chromatin structures from cells (p.7, l.44-16).

5. Page 6 lines 5-14. I feel the authors should discuss the possibility that direct nucleosome RNAPII contacts are established after elongation factors fall off. If there are technical reasons why they don’t think this is a possibility, they should address the issue.

Reply)

In the revised manuscript, we described a possible technical reason why the direct RNAPII-nucleosome contacts are mediated after the SPT4/5 and ELOF1 dissociation (p.6, l.25-28).

6. Discussion: the authors claim that they established a ChIP and cryo-EM method to solve endogenous structures. How does this approach differ from similar methods used to image chromatin bound complexes purified from cells?

Reply)

Thank you very much for this comment. The ChIP-CryoEM method can provide high-resolution (~3-4 Å) structural information of native target proteins bound to chromatin. Especially, this method facilitates the preparation and structural analysis of nucleosome-bound protein complexes, in which the genomic DNA is wrapped around histones. Chromatin is a nucleosome oligomer, and this oligomeric structure often forms aggregates, preventing high-resolution three-dimensional studies using single particle analysis. The ChIP-CryoEM technique overcomes this low-resolution problem and can reveal the mechanisms of protein functions in the native state of chromatin. In addition, the ChIP-CryoEM method provides the structural information of unexpected novel complexes containing a target protein. This study is the first determination of the three-dimensional structures of the ChIPed nucleosome-bound complex at 3-4 Å. We discussed these advantages of the ChIP-CryoEM method in the last paragraph of the Discussion (p.7, l.42-p.8, l3).

Language

A few sentences should be revised.

Abstract:

- Line 10: corresponding disassembly process. Please correct.
- Line 14 “during nucleosome reassembly process” should be “during nucleosome reassembly” or “during the nucleosome reassembly process”.
- Line 15 “transcribing human genome” should be “the human genome”.

Main text:

- Page 3 line 12: “two each of these histones” should be revised.
- Line 13 “in which” should be “around which”.
- Page 4 line 4. Associated and not associating.
- Page 5 line 6. I suggest “also represent an average of heterogeneous sequences”.

Reply)

We corrected these sentences, accordingly.

Reviewer #2 (Remarks to the Author):

Kujirai et al. report cryo-EM structures of RNA polymerase II (RNAPII) transcription elongation complexes (ECs) obtained by a novel immunoprecipitation method from nuclear extracts of live human tissue culture cells. This approach is important because it can give insight into the states of ECs and associated factors in living cells. As the first well-executed report of this approach, the manuscript is appropriate for high-visibility publication. ECs collected from living cells are expected to be in a variety of states because the EC associates with different factors as it transitions from transcript initiation to elongation through nucleosomes, mRNA splice sites, and polyA-addition sites, and then finally to a termination state after the mRNA is cleaved at the polyA site. The authors are able to identify four types of complexes by computational particle sorting of the heterogeneous EC mixture recovered by immunoprecipitation: (1) an EC containing only RNAPII, DNA, and RNA; (2) an EC containing elongation factors Spt4, Spt5, Spt6, and Elof; (3) an EC that has encountered a nucleosome and is halted at so-called nucleosome position SHL(-5); and (4) an EC that is exiting a nucleosome and is halted at so-called position SHL(+5). The major finding is simply that these complexes can be recovered by immunoprecipitation and analyzed by cryo-EM. This is an important advance because it validates decades of RNA polymerase structural analyses that have relied on purified proteins and reconstituted complexes. Although additional insights are somewhat limited, the overall story merits publication. That said, major revision is required. The authors suggest that the absence of elongation factors in all but the second complex indicates that ECs in vivo bind and release elongation factors as a function of template position or relationship to a nucleosome. Although possible, this conclusion is not supported by their findings because it is equally likely the factors dissociated during the course of sample preparation. A more balanced interpretation of their results will be required to merit peer-reviewed publication.

Reply)

Thank you very much for this comment. We revised the manuscript according to this reviewer's suggestions.

Major points.

1. As noted in the summary, the conclusion that some ECs, in particular those that are entering and departing a nucleosome, lack elongation factors is not supported by the authors results. The isolation of complexes involved multiple wash steps before crosslinking is applied to stabilize the complexes. There's nothing wrong with this approach, but it precludes conclusions about whether factors are present on the complexes at the point of nuclei isolation. The authors should discuss the potential impact of the isolation procedure in a more balanced fashion that makes explicit the inability to draw conclusions about the native state of the complexes.

Reply)

Thank you very much for this comment. We agree with this reviewer's opinion. Accordingly, in the revised manuscript, we toned down the discussion of the functional relevance of RNAPII without elongation factors, and intensively rewrote the possible dissociation of elongation factors during sample preparation in the revised manuscript (p.5 l.18-l.21, p.6 l.25-33, p.7 l.9-10, p.7 l.39-43).

2. The authors label their EC lacking elongation factors an 'RNAPII' complex rather than an elongation complex (EC) and then use the EC nomenclature to refer to the EC plus elongation factors. This choice directly contradicts over two decades of precedent in the scientific literature and the transcription field and will confuse readers. An EC is a complex of RNAPII, RNA, and DNA engaged in transcript elongation. The authors 'RNAPII' complex is an EC. RNAPII might be present in the sample, but the authors do not comment on whether or not they see evidence for RNA–DNA-free RNAPII (they might add this to the discussion). The authors should adopt a different nomenclature consistent with precedent and refer to all ECs as ECs. For example, they could use EC– and EC+ to indicate ECs with and without elongation factors, respectively (or any number of other alternatives).

Reply)

Thank you for this comment. In the revised manuscript, we changed the terminologies for RNAPII alone and EC to RNAPII and RNAPII-SPT4/5-ELOF1-SPT6, respectively.

3. The authors state that their complexes are pretranslocated. Their cryo-EM maps do not support this simplistic statement and rather suggest the complexes are mixtures of pre-translocated and post-translocated ECs as might be expected from a heterogeneous population of ECs isolated from cells. The cryo-EM signal for the modeled 3' RNA nucleotide in the pretranslocated register is significantly weaker than the signal for the penultimate RNA nucleotide. This is a clear indication that

some or even many of the ECs lack an RNA nucleotide is the product subsite of the active site and are post-translocated. The interpretation should be modified to accurately reflect the data by stating the complexes are mixtures of pre- and post-translocated ECs, with the fraction of each difficult to assess.

Reply)

Thank you for this insightful comment. In the revised manuscript, we corrected the corresponding sentence according to this suggestion with the new figures (p.5,l.15-18).

4. The authors rely on an antibody-targeted adapter insertion method they call ChIL-seq to map RNAPII and elongation factor association with DNA in HeLa cells. The authors argue that differences in the patterns for RNAPII ChIL peaks and Spt5–6 peaks indicates that RNAPII is sometimes localized to gene bodies without associated Spt5 and Spt6. These data and analyses are wholly inadequate to support such a conclusion. It's unclear how the histogram of peak distributions across a set of genes (Fig. 1e) even supports the conclusion since it doesn't directly depict the extent of peak co-localization. Beyond that, no statistics are offered to establish that the patterns differ. Genome-scale analyses without robust statistics are largely meaningless. In particular, triplicate datasets would be required and analyses provided showing the pattern differ more between RNAPII and Spt5–6 than between the replicates to statistically significant extent. Absent such analyses and a better illustration of the patterns of peaks observed, the results do not support a conclusion that RNAPII is bound in the body of genes without elongation factors. The single-molecule measurements of Spt5 association and disassociation cited by the authors do not address occupancy of ECs by Spt5 in vivo. This part of the manuscript should be improved or removed.

Reply)

We agree with this reviewer's opinion that RNAPII without elongation factors may be visualized, because the elongation factors were washed out during the sample preparation procedure. Therefore, in the revised manuscript, we intensively discussed the possible dissociation of elongation factors during sample preparation (p.5 l.18-1.21, p.6 l.25-33, p.7 l.9-10, p.7 l.39-43). We also understand the difficulty of using our genome-wide analysis data to conclude that RNAPII exists without elongation factors in the human genome, as this reviewer pointed out. Therefore, according to this reviewer's suggestion, we removed the data related to the genome-wide analysis in the revised manuscript.

5. The authors mention that previous genome-scale analyses have shown RNAPII paused at the –5 nucleosome position in vivo. Do the authors' Spt5–6 data (or other published Spt5–6 data) suggest Spt5–6 localize to the –5 nucleosome position or is it depleted from this location? The authors argue their structure of ECs lacking Spt5–6 at this location indicate that the nucleosome-paused EC lacks Spt5–6. The genome-scale data should be analyzed to directly test if this is true in vivo and the result reported with the authors findings.

Reply)

In the revised manuscript, we removed the data related to the genome-wide analysis.

6. Some features modeled into the provided pdb files do not appear to be supported by the cryo-EM maps. For example, the portion of the trigger loop near the rim helices and the RNA upstream of RNAPII interacting with Spt6, which are modeled in the EC plus elongation factor structure, do not appear to be supported by corresponding signal in the cryo-EM map. The authors should carefully curate the models for each of the structures they are reporting and remove from them any segments of polypeptide not supported by the cryo-EM data. Additionally, in regions of the models where resolution is clearly insufficient to support modeling of amino acid-side chains (e.g., Spt6 in the EC plus factors model), the authors should limit the model to the Ca backbone.

Reply)

We removed the trigger loop models in the PDBs. In our focused refine map of SPT6, the side chain densities of SPT6 were partially visualized. The RNA around SPT6 was also visualized. We present these maps in the revised manuscript (Fig. S5).

Minor points

1. The first paragraph of the results cites refs. 1–3 after each sentence. Citing once after the topic sentence is sufficient.

Reply)

We corrected the citations in the first paragraph of the Introduction, according to this suggestion.

2. line 131 and throughout manuscript. Results should be reported in past tense. "...all twelve protein subunits *were* observed..." (not "are observed").

Reply)

We corrected it accordingly.

3. line 218. The authors should explicitly state they are comparing structures of a yeast EC to the human EC results and give the citation to the yeast results at the end of the sentence.

Reply)

We corrected it accordingly.

4. Figs. 3b and 4b. Why do the arrows point in opposite directions in the 180° rotated

views? This difference will confuse readers trying to relate the two images to each other.

Reply)

We corrected these figures.

Reviewer #3 (Remarks to the Author):

Kujirai et al., present their findings using an innovative method to prepare RNA Polymerase II (RNAPII) samples for cryo-EM analysis directly from HeLa cells: ChIP-Cryo-EM. Their approach takes advantage of the ability to directly isolate FLAG tagged RNAPII from HeLa cells, followed by a series of washes, sample separation and crosslinking via GraFix, and analysis using single-particle cryo-EM. This methodology successfully identified several RNAPII complexes actively engaged in transcription elongation. The authors describe two RNAPII elongation complexes, one in the absence and one in the presence of elongation factors. Additionally, they report two transcription elongation complexes lacking transcription factors but featuring a nucleosome either upstream or downstream of RNAPII. They suggest that these RNAPII-nucleosome conformations resemble the chromatin disassembly and reassembly processes during transcription.

This manuscript offers a novel approach to capturing transcription complexes that complements traditional in vitro reconstitution methods. Their technique, as described, serves as a proof of concept. However, some key controls are missing, and a few of the claims, particularly the role of an RNAPII elongation complex without elongation factors as a key intermediate of gene-body transcription, are not fully supported by the presented data. Specific comments the authors should address prior to publication are provided below.

Major Comments:

1. ChIL-seq Experiment: The authors use this assay to support their claim that RNAPII is actively transcribing coding sequences without DSIF and SPT6. Several issues complicate this interpretation, and the methodology is described in a way that makes it difficult to follow:

- o It appears that all three antibodies (anti-FLAG for RNAPII, -SPT5, and -SPT6) were added simultaneously. The varying affinity of each antibody could impact the efficiency of capturing all three proteins when used together. No catalog numbers are provided for the antibodies, making it impossible to evaluate their suitability for this experiment.

- o The presence of RNAPII elongation complexes is inferred by the colocalization of FLAG, SPT5, and SPT6 ChIL-seq peaks. If FLAG peaks do not overlap with the others, the authors consider this an RNAPII elongation complex without elongation factors. However, RNAPII elongation complexes could lack DSIF or SPT6, and instead include factors involved in transcription coupled repair, the super elongation complex, elongin, NELF, or RNA processing complexes. Additionally, elongation factors like SPT5 can exhibit flexibility without disrupting the entire complex (see further comments on this in point 3 below). Such flexibility could reduce the efficiency of the secondary antibody to insert the transposable sequence near chromatin,

especially in nucleosome-rich regions, complicating the interpretation of the results. The authors should consider alternative explanations for the absence of peaks.

o The secondary antibody's transposon insertion efficiency and library preparations may introduce biases in peak identification. This issue is not addressed in the experimental design, and these biases could obscure the conclusion that RNAPII alone is transcribing. The authors should discuss these potential biases.

o How do these observations compare with ChIP-seq experiments? Previous ChIP-seq data show continuous SPT5 and SPT6 occupancy throughout gene bodies, and, as far as I am aware, there is no evidence of depletion of these factors in the vicinity of nucleosomes.

o Finally, the claim that a substantial portion of RNAPII elongation complexes would transcribe through chromatin in the absence of any elongation factors conflicts with numerous studies. It is known that RNAPII elongation complexes are slow and inefficiently overcome the nucleosome barrier without elongation factors in vitro and in cells. The assertion that RNAPII elongation complexes lacking elongation factors represents a physiological state requires more substantial evidence and would be a major deviation from our current understanding of transcription elongation in mammals.

Reply)

Thank you very much for your thoughtful comments on our genome-wide analysis. Your feedback has given us the opportunity to explore various possibilities and reevaluate our structural data from a more balanced perspective. In particular, Reviewers 1 and 2 raised important points regarding the potential dissociation of elongation factors during sample preparation. We agree that confirming the presence of RNAPII without elongation factors in the human genome is indeed challenging with current techniques. Additionally, we acknowledge that elongation factors may dissociate during sample preparation. As a result, in the revised manuscript, we have removed the genome-wide analysis and substantially revised the discussion to reflect the possibility that RNAPII without elongation factors may arise from their dissociation during sample preparation, while also considering the potential existence of this form (p.5 l.18-l.21, p.6 l.25-33, p.7 l.9-10, p.7 l.39-43).

2. ChIP-Cryo-EM:

o Although the authors mention sample fractionation by GraFix, it is unclear whether each fraction was individually frozen on grids. If so, it would be important to note whether the fractions contained different complexes. For example, if the large complexes the authors mentioned at the end of the discussion are enriched in certain fractions, they would be omitted from the data collection. This could include complexes representing initiation complexes, complexes bound to PAF1c, or RNA processing machineries. It appears that the complexes that are presented here would have a molecular weight < 1 MDa, suggesting that larger complexes are lost during the GraFix purification.

Reply)

According to this suggestion, we performed the sucrose gradient ultracentrifugation under the same conditions as in GraFix, but without glutaraldehyde. We then

analyzed the protein composition in these fractions, including the corresponding fractions subjected to the cryo-EM analysis, by western blot. We found that the collected fractions mainly contained the RNAPII elongation complex. The preinitiation complex components, XPB and MED17, were enriched at the bottom of the gradient. Therefore, the preinitiation complex may have been removed during GraFix. These new data are presented in the new Fig. 1b and the results are described in the text (p. 4 l.31-p.5 l.6).

o Including a denaturing gel of non-crosslinked GraFix fractions (running a second sample through the sucrose gradient without glutaraldehyde) would help determine the subunit composition of the samples. Additionally, mass spectrometry of each GraFix fraction would clarify the proportion of additional factors associated with RNAPII.

Reply)

As mentioned above, we performed the sucrose gradient ultracentrifugation under the same conditions as in GraFix, but without glutaraldehyde, and analyzed the protein composition in these fractions by western blot analysis. These new data clarified the proportions of factors associated with RNAPII (new Fig. 1b), and the results are described in the text (p.4 l.41-p.5 l.6).

o The authors mention the potential loss of factors during the washing steps, but no controls are provided to demonstrate that these factors are indeed lost in the washes. Additionally, phosphatase inhibitors only appear to be included in the initial wash and not included in later incubation steps. It is known that RNAPII phosphorylation state can greatly affect the stability of the elongation factors that associate with the RNAPII elongation complex. The authors use 150 mM NaCl in their early wash steps, and this salt concentration would cause more weakly associated elongation factors like RTF1 to disassociate from RNAPII. It would be worth mentioning if the authors have tried less stringent washing procedures and suggest this as an additional parameter that can be manipulated in future iterations of this experiment.

Reply)

Thank you very much for this insightful suggestion. As this reviewer suggested, we performed the sample preparation under less stringent conditions (50 mM NaCl). We then compared the remaining elongation factors with those obtained under the 150 mM NaCl conditions used for the ChIP-CryoEM analysis. As the reviewer suggested, we found that that a relatively small amount of PAF1 was associated with RNAPII, and RTF1 was only detected when the sample was prepared under low salt conditions (50 mM). The phosphorylation of Serine 2 was confirmed. This information is provided in the new Supplementary Fig. 1f, and described in the Results section (p.4 l.31-l.41).

3. The authors provide extensive explanations for the interactions between DSIF and ELOF1 and SPT6 and DSIF. In these discussions, they do not mention the KOW5 domain of DSIF. This is the most stably bound portion of DSIF on RNAPII and

appears to be present in the provided models. The authors state the DSIF is not present in the nucleosome structures. Have they carefully examined their maps for density corresponding to SPT5 KOW5? The reviewer acknowledges that the other observed DSIF domains would clash with the position of the upstream nucleosome, and these domains would be flexibly tethered to RNAPII by the KOW5 domain.

Reply)

Thank you for this comment. In the RNAPII-SPT4/5-ELOF1-SPT6 complex structure, the SPT5 KOW5 domain was visualized. We added a description of the KOW5 structure in the manuscript (new Fig. 2e, p.5 l.31-32). In the RNAPII-upstream nucleosome, we confirmed again that the KOW5 domain was not visualized.

4. The authors need to adjust their terminology for the complexes they are studying to be consistent with the literature. All complexes the authors investigate structurally are transcription elongation complexes. The description of elongation complexes lacking elongation factors should not be described as “transcribing RNAPII” or “RNAPII alone”. To make it clear, they should call this complex an RNAPII elongation complex. For their other elongation complex with factors, it would be best to call this a RNAPII-ELOF1-DSIF-SPT6 elongation complex. Similarly, use of @DS or @US to describe the location of the nucleosome relative to RNAPII is not intuitive. Here it would be best to describe this as an RNAPII elongation complex associated with an upstream nucleosome (or downstream nucleosome). The use of abbreviations can make it difficult for others in the field and outside of the field to understand the exact nature of the complexes are being studied.

Reply)

Thank you very much for this suggestion. In the revised manuscript, we changed the terminologies for RNAPII alone and EC to RNAPII and RNAPII-SPT4/5-ELOF1-SPT6, respectively. In addition, as suggested by this reviewer, we also changed the terminologies for @US and @DS to RNAPII associated with the upstream nucleosome and RNAPII associated with the downstream nucleosome, respectively.

5. Line 195-196- The authors state that the RNAPII upstream nucleosome complex contains “tightly rewrapped” DNA around the nucleosome. They have no evidence to indicate that this nucleosome was actually transcribed in their experiments. It would be best to state that the “DNA is tightly wrapped” to avoid overinterpretation of their data.

Reply)

We removed “tightly” from the corresponding sentence.

6. Line 73- the authors state “before the center of the nucleosomal DNA”. They should call this the nucleosomal dyad.

Reply)

We corrected it accordingly.

7. Line 45- The first sentence of the introduction is misleading. RNAPII transcribes non-coding and protein coding regions of the genome, but other RNAPs also can perform these functions. They should rephrase to emphasize the RNAPII primarily transcribes protein coding genes and some non-coding gene regions.

Reply)

We corrected it accordingly.

8. Line 47-The authors state “During gene expression, the storage RNAPII molecule”. I am not sure what the “storage RNAPII molecule” is. Do they mean RNAPII dimers or RNAPII bound to RPAP2? To avoid confusion, I would simplify this sentence to state RNAPII is assembled on promoters.

Reply)

We corrected it accordingly.

9. The paragraphs starting at lines 257 and 269 respectively, could be swapped to more accurately follow the flow of the presented text.

Reply)

We corrected it accordingly.

10. The authors state that they do not observe PAF1c bound to RNAPII in their cryo-EM data. Have they tried to further classify their SPT6 particles? PAF1c is difficult to initially observe in cryo-EM and they may be model biased in their classification approaches. They have 130k particles for the SPT6 EC, and there is a possibility that a fraction of these particles have weak PAF1c density. Also, have the authors looked for SPT6 tSH2 density in their maps using focused classification approaches?

Reply)

As suggested by this reviewer, we have tried to visualize PAF1C and the tSH2 domain of SPT6. However, we could not observe these factors and domain in the current analysis. The classification results are attached here. In the new western blot (comment 2), only a small amount of PAF1C was detected in our sample prepared for cryo-EM analysis. We think the absence of the PAF1C structure is consistent with this western blot analysis and the previous reports that PAF1C is transiently associated with the RNAPII elongation complex (Fischl et al., Mol Cell, 2017; Versluis et al., Mol Cell, 2024).

- (a) 3D classification of RNAPII-SPT4/5-ELOF1-SPT6. PAF1C was not visualized after 3D classification of RNAPII-SPT4/5-ELOF1-SPT6 particles.
- (b) RNAPII-SPT4/5-ELOF1-SPT6 map (yellow) with PDB:6GMH. tSH domain (dotted circle) was not visualized in the map at low contour level.
- (c) tSH domain was not visualized after focused 3D classification of RNAPII-SPT4/5-ELOF1-SPT6 particles.

Minor Comments

- Introduction (p. 3, lines 29–38): The description of nucleosome transcription by RNAPII is unclear. It is particularly difficult to distinguish which findings are specific to yeast and which apply to metazoans. The authors should also include reference 50 in their introduction for prior structural work.

Reply)

At the end of the fourth paragraph of the Introduction, we now extensively describe an alternative histone relocation pathway with human and *K. pastoris* RNAPIIs in the histone chaperone-independent manner, with the new citations suggested by this reviewer (p.2 l.40-45).

- Introduction (p. 3, line 39 onward): The authors describe the complexity of factors involved in chromatin transcription. However, their findings suggest the opposite, that the RNAPII elongation complex is present near nucleosomes without elongation factors, which conflicts with the literature they cite. The authors should address this apparent contradiction, possibly in the discussion section.

Reply)

As this reviewer pointed out, our findings about the RNAPII-nucleosome complexes from human cells may seem to contradict the previous findings. This can be reconciled if we discuss the possible dissociation of the elongation factors and the repositioning of the nucleosome during sample preparation. Accordingly, we extensively rewrote these parts throughout the revised manuscript.

- Figure 1D and 1E: The provided graphics are difficult to interpret. It is unclear whether the counts are normalized against the number of peaks or another criterion. If there are more peaks classified as RNAPII alone, it would be expected to see more counts along the coding regions. The presentation of the data is unclear. Adding example gene tracks and correlating the data with nucleosome positioning from previously published MNase-seq could provide better context for interpreting the peaks.

Reply)

We agree with all three reviewers' concerns that elongation factors may dissociate during the sample preparation. We also agree that confirming the presence of RNAPII without elongation factors in the human genome is indeed challenging with current techniques. Therefore, in the revised manuscript, we have removed the genome-wide analysis and substantially revised the discussion to reflect the possibility that RNAPII without elongation factors may arise from their dissociation during sample preparation, while also considering the potential existence of this form.

- Discussion (p. 7, lines 26–36): The authors cite Rosen G.A. et al (reference 40). to support their argument but do not address conflicting findings that demonstrate the essential roles of SPT5, SPT6, and PAF1c in transcription elongation. How can elongation factors dissociate without being outcompeted by other binding factors in a cellular context? The only referenced previous work was done with yeast components for a single molecule analysis. Is there any prior evidence of such dynamic binding of DSIF for the human RNAPII?

Reply)

Thank you very much for this comment. We agree that our previous discussion may conflict with the previous report, demonstrating the essential roles of SPT5, SPT6, and PAF1C. To reconcile this, in the revised manuscript, we state that RNAPII without elongation factors may be visualized because the elongation factors dissociated during the sample preparation procedure. We intensively discussed the possible dissociation of elongation factors during sample preparation in the revised manuscript (p.5 l.18-l.21, p.6 l.25-33, p.7 l.9-10, p.7 l.39-43).

- Figure 2a: What do the dotted lines represent? Are these the regions that the authors observe in the structure? If so, KOW6-7 has not been visualized in a transcription EC to date (and is included in this box) and the SPT6 tSH2 has been visualized in multiple structures (and is excluded from this box).

Reply)

The dotted lines represented DSIF, but it is confusing. We corrected the figure in the revised manuscript.

- 2D class images: Scale bar is missing. (Fig S2b)

Reply)

We added the scale bar.

- Supplementary figure 8: Both of these active sites represent elongation complexes. (related to point 4)

Reply)

We corrected the figure.

REVIEWER COMMENTS

Reviewer #1 (Remarks to the Author):
Comment)

The authors addressed all the points I raised. I feel this manuscript is now ready for publication.

Reply)

Thank you very much. I appreciate it very much.

Reviewer #2 (Remarks to the Author):
Comment)

The authors have substantially revised the manuscript to address the concerns of the reviews. Importantly, the CHIP-seq data, which had been over-interpreted and did not support the conclusions drawn, is removed from the revised manuscript. Further, the statements about ECs lacking elongation factors have been revised to acknowledge that the factors may well have dissociated during isolation of complexes. A number of other revisions have improved the interpretation of the data significantly. As revised, the manuscript now reports an important advance in the elucidation of RNAPII structures from cells. The revised manuscript is largely suitable for publication, but there is still one issue of nomenclature that the authors should fix. Reviewers 2 and 3 both asked that the authors not refer to elongation complexes as RNAPII but as elongation complexes. This distinction is important because RNAPII can exist in many forms, including not bound in active transcription complexes but the authors report exclusively on elongating transcription complexes. References to RNAPII may confuse readers by implying RNA and DNA may not be present in the complexes. Although acknowledging the issue and making some revisions, the confusion still persists in the revised manuscript. This may simply be a matter of translation to English. I think using the EC abbreviation for elongating or elongation complex that was established in the first report of RNAPII EC structures (Gnatt et al., 2001. *Science* 292:1876) would make the manuscript easier for a general audience to follow. To illustrate this point, I suggest revising the paragraph in the results starting at line 138 to read as follows. This nomenclature could then be used thereafter in the rest of the manuscript.

"We first obtained two distinct native RNAPII elongation complex (EC) structures, EC with RNAPII alone and EC with RNAPII complexed with SPT4/5, ELOF1, and SPT6 at 2.7 Å and 3.1 Å resolutions, respectively (Fig. 1c, d). These ECs were actively transcribing genomic DNA or stalled on certain genomic regions in human cells. In both EC structures, all twelve RNAPII subunits were observed (Fig. 1c, d). Genomic DNA and nascent RNA fragments were clearly visualized around the RNAPII catalytic center, suggesting that they were actively transcribing forms of human RNAPII ECs (Supplementary Fig. 8a,d). Since these are average structures of ECs at various genomic DNA loci, the cryo-EM maps of the DNA and RNA bases represent an average of heterogeneous sequences. Further, these ECs may be mixtures of pre-translocated and post-translocated forms because the densities of the RNA base at the +1 position

are relatively weak (Supplementary Fig. 8b,c,e,f). ECs without elongation factors may be observed as a consequence of elongation factor dissociation during sample preparation for the cryo-EM analysis. Alternatively, ECs without elongation factors may function in genome transcription at certain loci. Further studies are required to solve this issue.”

Reply)

Thank you very much. We revised the manuscript accordingly.

Reviewer #3 (Remarks to the Author):

General comment)

The authors have mostly addressed my previous concerns. Here are a few additional points that should be addressed prior to publication.

Reply)

Thank you very much for these additional comments. We revised our manuscript accordingly.

Comment 1)

Lines 166-168 and line 299-300— It is odd the the authors compare their structure to the reconstituted yeast complex when there is a reconstituted mammalian complex, which would more closely resemble what they are studying here. The authors should minimally cite DOI: 10.1038/s41586-018-0440-4

Reply)

For lines 166-168, we compared SPT6 in mammalian EC structures and found that, in the suggested study (PDBs: 6GMH and 6TED), SPT6 is partially detached from SPT5. In contrast, SPT6 binding in our structure aligns with the recently published reconstituted mammalian EC-SPT4/5-IWS1-PAF1C-SPT6-SETD2 structure (DOI: 10.1126/science.adn6319). This difference likely reflects the flexibility of SPT6 binding. We have described this comparison in the manuscript (p.5, l.43–p.6, l.3) and in the newly added Supplementary Fig.9.

For lines 299-300, our sentence describes ELOF1 potentially functioning with SPT4/5 and SPT6. However, the suggested study (DOI: 10.1038/s41586-018-0440-4) does not include ELOF1, and no published mammalian EC structures contain ELOF1 in complex with SPT4/5 and SPT6. Therefore, we referenced the yeast structure in this context.

Comment 2)

Paragraph starting at line 238- The authors again compare their human structures to a *K. pastoris* structure. There are now multiple structures of upstream NCPs on the mammalian EC, and these would be a more appropriate for comparison (can be added in addition to the *K. pastoris* data). See DOIs: 10.1126/science.adn6319 and 10.1101/2024.12.13.628244

Reply)

In this section, we compared the interaction between nucleosomal H2A-H2B and EC in structures where the nucleosome is nearly fully reassembled (~120 bp DNA rewrapped). The mammalian EC structures with upstream nucleosome-FACT or SETD2, as suggested by the reviewer, do not contain a nearly reassembled nucleosome but rather represent either an early stage of nucleosome reassembly (FACT-associated) or a post-reassembly state (SETD2-associated). These mammalian structures do not exhibit interactions between H2A-H2B and EC. Therefore, a direct comparison between our human EC-upstream nucleosome and these mammalian EC-upstream nucleosome-FACT (or SETD2) structures would not be appropriate.

Comment 3)

Supplementary Figure 8-

Panel G is cited, and there is no panel G. It appears that panel D was omitted from the figure legend.

Thank you for this comment. We corrected the legend accordingly.

REVIEWERS' COMMENTS

Reviewer #2 (Remarks to the Author):
Comment)

The authors have satisfactorily revised the manuscript and it is ready for publication.

Reply)

Thank you very much.

Reviewer #3 (Remarks to the Author):
Comment)

The authors have addressed my comments.

Reply)

Thank you very much.